# Opposite changes in APP processing and human Aβ levels in rats carrying either a protective or a pathogenic APP mutation

Marc D Tambini, Kelly A Norris, Luciano D'Adamio*

Department of Pharmacology Physiology & Neuroscience New Jersey Medical School, Brain Health Institute, Jacqueline Krieger Klein Center in Alzheimer's Disease and Neurodegeneration Research, Rutgers, The State University of New Jersey, Newark, United States

**Abstract** Cleavage of APP by BACE1/β-secretase initiates the amyloidogenic cascade leading to Amyloid-β (Aβ) production. α-Secretase initiates the non-amyloidogenic pathway preventing Aβ production. Several *APP* mutations cause familial Alzheimer's disease (AD), while the Icelandic *APP* mutation near the BACE1-cleavage site protects from sporadic dementia, emphasizing APP's role in dementia pathogenesis. To study APP protective/pathogenic mechanisms, we generated knock-in rats carrying either the protective (*App^P*) or the pathogenic Swedish mutation (*App^s*), also located near the BACE1-cleavage site. α-Cleavage is favored over β-processing in *App^P* rats. Consequently, non-amyloidogenic and amyloidogenic APP metabolites are increased and decreased, respectively. The reverse APP processing shift occurs in *App^s* rats. These opposite effects on APP β/α-processing suggest that protection from and pathogenesis of dementia depend upon combinatorial and opposite alterations in APP metabolism rather than simply on Aβ levels. The Icelandic mutation also protects from aging-dependent cognitive decline, suggesting that similar mechanisms underlie physiological cognitive aging.

*For correspondence:
luciano.dadamio@rutgers.edu

Competing interests: The authors declare that no competing interests exist.

## Introduction

The Amyloid Precursor Protein (APP), the mutation of which can cause or prevent Alzheimer's disease (AD), is an extensively processed type 1 transmembrane protein. The β-processing of APP involves an initial cleavage of the APP ectodomain by β-secretase (BACE1), followed by the cleavage of the resultant β-COOH-terminal fragment (βCTF) by γ-secretase to produce amyloid beta (Aβ) and the APP intracellular domain (AID/AICD). Alternatively, in the nonamyloidogenic pathway, APP is first cleaved within the Aβ region sequentially by α- and γ-secretase, which produces a smaller P3 fragment and AID/AICD (*Sisodia et al., 2001*; *Sisodia and St George-Hyslop, 2002*). APP processing is central to understanding AD for two reasons: 1. The biological functions of APP and its metabolites affect many important neuronal functions, related to AD (*Barbagallo et al., 2011*; *Barbagallo et al., 2010*; *Guo et al., 2012*; *Matrone et al., 2011*; *Matrone et al., 2012*). Not just amyloid beta, but the other APP metabolites and the holoprotein itself have important effects on the neuron (*Del Prete et al., 2014*; *Fanutza et al., 2015*; *Fogel et al., 2014*; *Gulisano et al., 2019*; *Matrone et al., 2011*; *Nikolaev et al., 2009*; *Passer et al., 2000*; *Puzzo et al., 2017*; *Rice et al., 2019*; *Tambini et al., 2019*; *Yao et al., 2019*; *Zott et al., 2019*). 2. Mutations that affect APP processing either cause or prevent AD. This is true for both early and late-onset AD (*Zhang et al., 2011*). For example, the K670N/M671L (Swedish) mutation of the NH2-terminal side of the β-cleavage site of APP increases its β-processing and causes a familial form of AD (*Citron et al., 1992*; *Citron et al., 1994*; *Johnston et al., 1994*). The A673T (Icelandic or protective) mutation, which is COOH-terminal to the β-site of APP, decreases the affinity of APP for BACE1, and thereby reduces β-processing and

protects against late-onset AD and normal cognitive decline (*Jonsson et al., 2012*). Late-onset AD can also be caused by mutations which block the α-processing of APP (*Hartl et al., 2018*; *Suh et al., 2013*).

The choice of animal model and genetic approach has important implications for the study of the effects of mutations on APP processing. Analysis of transgenic APP models, which are extensively used in AD research, are confounded by several factors: the overexpression of APP above physiological levels (*Saito et al., 2016*), the disruption of genes in the transgene integration sites (*Goodwin et al., 2019*; *Tosh et al., 2017*), the use of exogenous promoters, which do not replicate the temporal, cell type-specific or spatial expression of the endogenous gene (*Rodgers et al., 2012*), and the overproduction of multiple, biologically active APP metabolites. This last point is potentially important: the reduction in some APP metabolites, due to either pathogenic or protective mutations, may participate in either pathogenic or protective mechanisms. A transgenic approach will not mimic this loss of function effect but, on the contrary, it will generate a gain of function model organism (i.e. the opposite of the human process that the organism aims to model). A knock-in (KI) approach, in which the mutations of interest are introduced into the endogenous gene locus and in which the endogenous regulatory elements of the gene are left intact, eliminates these confounding factors.

Rats are better suited to study neurodegenerative diseases for the following reasons. The rat was the organism of choice for most behavioral, memory and cognitive research, which is critical when studying neurodegenerative diseases, because physiological processes are similar in rats and humans, and the rat is an intelligent and quick learner (*Deacon, 2006*; *Foote and Crystal, 2007*; *Kepecs et al., 2008*; *Whishaw et al., 2001*). Several procedures that are important in dementia research are more easily performed in rats as compared to mice due to the larger size of the rat brain. Cannulas -to administer drugs, biologics, viruses etc.- and micro-dialysis probes –for sampling extracellular brain levels of neurotransmitters, Aβ, soluble tau etc.- can be accurately directed to individual brain regions, causing less damage and increasing specificity. In vivo brain imaging techniques, such as MRI (*Bartelle et al., 2016*) and PET (*Leuzy et al., 2014*; *Zimmer et al., 2014a*; *Zimmer et al., 2014b*), can assess the extent and course of neurodegeneration with better spatial resolution in rats. Moreover, rats are large enough for convenient in vivo electrophysiological recordings or serial sampling of cerebrospinal fluid for detection of biomarkers. Gene-expression differences suggest that rats may be advantageous model of neurodegenerative diseases compared to mice. Alternative spicing of *tau* (*Andreadis, 2005*; *Hong et al., 1998*; *Janke et al., 1999*; *Roberson et al., 2007*), which forms NFTs and is mutated in Frontotemporal Dementia (*Goedert et al., 1998*; *Grover et al., 2003*; *Grundke-Iqbal et al., 1986*; *Hutton et al., 1998*; *Kowalska et al., 2002*; *Spillantini and Goedert, 1998*; *Stanford et al., 2003*; *Yasuda et al., 2000*), leads to expression of tau isoforms with three or four microtubule binding domains (3R and 4R, respectively). Adult human and rat brains express both 3R and 4R tau isoforms (*Hanes et al., 2009*): in contrast, adult mouse brains express only 4R tau (*McMillan et al., 2008*), suggesting that the rat may be a better model organism for dementias with tauopathy.

Recent developments in gene-editing technologies can make the rat once more the organism of choice to study dementias. Thus, we used CRISPR/Cas9-mediated genome editing to create KI rat models of protective and pathogenic *APP* mutations. Rat and human APP differ by 3 amino acids in the Aβ region (*Figure 1A*): since aggregated forms of Aβ are by and large considered the main pathogenic molecule in AD and given that human Aβ may have higher propensity to form toxic Aβ species as compared to rodent Aβ, together with the protective and Swedish mutations we introduced mutations to 'humanize' the rat Aβ sequence. These rats are referred to as $App^p$ (protective) and $App^s$ (Swedish) rats. As controls, we produced rats carrying only the humanized Aβ sequence ($App^h$ rats) (*Tambini et al., 2019*).

The customary approach to study AD pathogenesis is to determine the mechanisms causing neurodegeneration in disease model organisms, such as the $App^s$ rats. The $App^p$ rats add a complementary approach aimed to determine the mechanisms by which the protective *APP* variant prevents age-associated cognitive decline and AD. Here, we present an analysis of $App^p$ rats, centered on the effects of this variant on APP metabolism. We compare these metabolic changes to those caused by the Swedish pathogenic mutation. Finally, we tested how the Swedish and protective APP mutants interact in vivo to determine levels of APP metabolism.

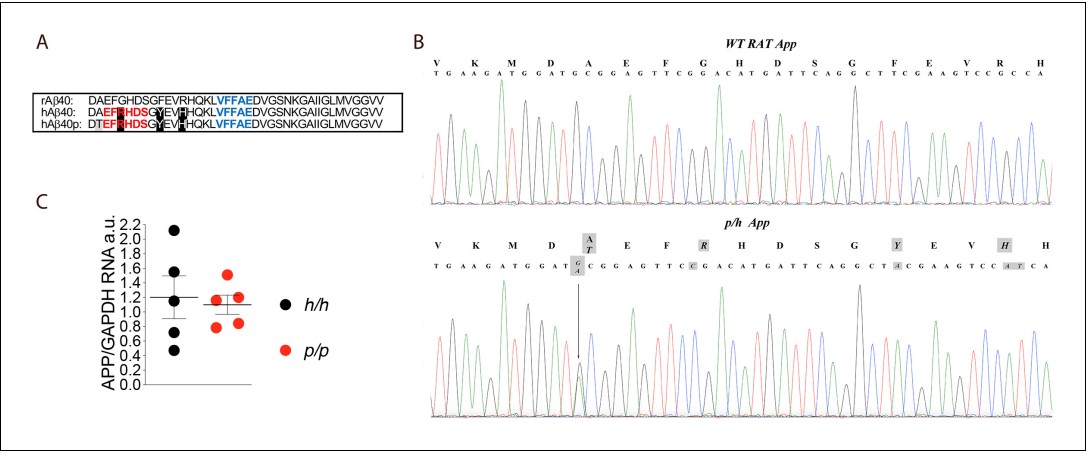

**Figure 1.** *App* is mutated to contain protective and humanizing mutations and *App* mRNA expression is normal in *App*$^P$ KI rats. (**A**) Alignment of Aβ40 region from wildtype rats (top), *App*$^h$ rats (middle), and *App*$^P$ rats (bottom). 6E10 epitopes are in red, 4G8 epitopes are in blue. Humanizing mutations are in highlighted in black. (**B**) PCR amplification of *App* gene exon-16 from *App*$^{w/w}$ and *App*$^{p/h}$ rats and sequencing of PCR product shows that the humanizing (G to C, T to A and GC to AT substitution) and protective (the G to A substitution) mutations were correctly inserted. Substituted nucleotides are highlighted in gray. The amino acid substitutions introduced by the mutations are highlighted in gray above the DNA sequences (G to T = A to T; G to C = G to R; T to A = Y to F; and GC to AT = R to H). (**C**) Levels of *App* mRNA were measured at p21 and normalized to *Gapdh* mRNA expression. We used the following male and female animals: *App*$^{h/h}$, 3 males and 2 females; *App*$^{p/p}$, 3 males and 2 females. Data were analyzed by unpaired student's t-test, and presented as average (*App*/*Gapdh*)± SEM. The online version of this article includes the following source data for figure 1:

**Source data 1.** Related to *Figure 1C*.

# Results

## *App* mRNA expression is normal in *App*$^P$ rats

The founder rat (F0#88, generated as described in the Materials and methods section) carrying the *p* and humanizing mutations was crossed to WT (*App*$^{w/w}$) Long-Evans rats to generate F1-*App*$^{p/w}$ rats. F1-*App*$^{p/w}$ rats were crossed to WT Long-Evans to generate F2-*App*$^{p/w}$ rats. To reduce to probability that F5 rats carry unidentified off-target mutations (except those, if present, on Chr. 11) to ~1.5625%, this crossing was repeated three more times to obtain F5-*App*$^{p/w}$ rats. Male and female *App*$^{p/h}$ rats were crossed to generate *App*$^{h/h}$, *App*$^{p/h}$ and *App*$^{p/p}$ animals. The humanizing and protective Icelandic mutations (*Figure 1A*) were correctly inserted into the *App*$^{p/h}$ genome (*Figure 1B*) and expression of *App* in *App*$^{p/p}$ brains was comparable to that detected in *App*$^{h/h}$ brains (*Figure 1C*, p=0.7576). We have previously shown that *App* mRNA levels in *App*$^{h/h}$ brains was identical to those observed in WT (*App*$^{w/w}$) Long-Evans rats (*Tambini et al., 2019*). Thus, the introduced humanizing plus protective Icelandic mutations do not alter *App* mRNA expression.

## Gene-dosage-dependent increased α-processing and reduced β-cleavage of APP in *App*$^P$ Knock In rats

APP is cleaved by several proteases. The most studied APP processing pathways involve APP cleavage by α-, β- and γ-secretases. Cleavage of APP by β-secretase takes place predominantly in acidic compartments: this processing releases a long (595 amino acid long, if referring to the APP695 isoform- the most highly expressed in brain) soluble NH$_2$-terminal APP ectodomain (sAPPβ) and a membrane-bound 99 amino acid long fragment (C99 or βCTF). βCTF is cleaved with lax site specificity by γ-secretase into Aβ peptide and the APP intracellular domain (AICD/AID). Alternatively, α-secretase cleaves APP along the secretory pathway or on the plasma membrane. This cleavage occurs within the Aβ sequence to produce soluble APPα (sAPPα) and a COOH-terminal membrane-bound 83 amino acid long fragment (C83 or αCTF). αCTF can be also cleaved by γ-secretase into a 'shorter Aβ' peptide, called P3 (see Figure 3A for a depiction of the major Aβ and P3 species), and AID.

The steady-state levels of APP metabolites in the brain depend on the rate of production, aggregation and catabolism. sAPPα and sAPPβ are stable APP metabolites (*Morales-Corraliza et al., 2009*), unlike αCTF and βCTF, which are cleared by γ-secretase and lysosomes (*Barbagallo et al., 2010*). Thus, sAPPα and sAPPβ are better indicators of the rate of cleavage of APP by either α- or β-secretase, respectively. As noted above, in vitro studies have shown that the protective Icelandic mutation reduces the rate of APP processing by β-secretase; in addition, a non-significant trend towards an increase in the rate of APP cleavage by α-secretase was also noted (*Jonsson et al., 2012*). To test whether these APP metabolic changes are reproduced in our $App^p$ KI rats and to verify whether the protein product of the $App^p$ allele contains the humanizing mutations, we analyzed brain samples isolated from $App^{w/w}$, $App^{\delta7/\delta7}$ -rats that contain two hypomorphic $App^{\delta7}$ alleles (*Tambini et al., 2019*), $App^{h/h}$, $App^{s/s}$, and $App^{p/p}$ rats. Young rats were chosen to avoid the confounding effect of amyloid aggregation, though it should be noted that no plaques were evident in 3 month old $App^{s/s}$ rats (*Tambini et al., 2019*). $App^{w/w}$ rats express unmodified rodent APP (i.e. containing the rat Aβ sequence); $App^{h/h}$ rats express endogenous rodent APP carrying the humanizing mutations (i.e. containing the human Aβ sequence) (*Tambini et al., 2019*); $App^{s/s}$ rats express endogenous rodent APP but with the humanizing mutations and the Swedish pathogenic mutations (*Tambini et al., 2019*); and $App^{p/p}$ rats express endogenous rodent APP carrying the humanizing mutations and the Icelandic protective mutation. Brain samples from these rat lines were tested by western blot (WB) with the following anti-APP antibodies. Y188, a rabbit polyclonal raised against the COOH-terminal 20 amino acids of APP, an epitope that is unchanged by the humanizing, Swedish and Icelandic mutations. M3.2, a mouse monoclonal raised against the rat APP sequence between the β- and α-secretase cleavage sites (DAEFGHDSGFEVRHQK); this antibody only recognizes APP molecules containing the rat Aβ sequence and not APP molecules containing the human Aβ sequence (DAEFRHDSGYEVHHQK, the 3 amino acid differences with the rat sequence are highlighted in black in *Figure 1A*). Conversely, 6E10, a mouse monoclonal whose epitope is shown in *Figure 1A*, only recognizes APP molecules containing the human Aβ sequence. Y188 detects APP, βCTF and αCTF in $App^{w/w}$, $App^{h/h}$, $App^{s/s}$ and $App^{p/p}$ brains (*Figure 2A*). M3.2 detects APP and βCTF only in $App^{w/w}$ brains (*Figure 2B*). On the other hand, 6E10 detected APP and βCTF only in $App^{h/h}$, $App^{s/s}$ and $App^{p/p}$ rats (*Figure 2C*). As expected, none of these antibodies gave specific signal in $App^{\delta7/\delta7}$ samples. Thus, APPh, APPSw and APPp, the protein products of the $App^h$, $App^s$ and $App^p$ alleles, contain the humanized Aβ sequence, respectively. M3.2 is specific for APP molecules containing the rat Aβ sequence; conversely, 6E10 only recognizes APP molecules containing the human Aβ sequence. Y188 detects APP, βCTF and αCTF in $App^{w/w}$, $App^{h/h}$, $App^{s/s}$ and $App^{p/p}$ brains (*Figure 2A*). M3.2 detects APP and βCTF only in $App^{w/w}$ brains (*Figure 2B*). On the other hand, 6E10 detected APP and βCTF only in $App^{h/h}$, $App^{s/s}$ and $App^{p/p}$ rats (*Figure 2C*). As expected, none of these antibodies gave specific signal in $App^{\delta7/\delta7}$ samples. Thus, APPh, APPSw and APPp, the protein products of the $App^h$, $App^s$ and $App^p$ alleles, contain the humanized Aβ sequence.

Next, we used an antibody raised against the COOH-terminus of human sAPPβ and an antibody raised against the COOH-terminus of sAPPα to perform WB analysis on soluble brain fractions. sAPPβ was significantly lower in $App^{p/p}$ as compared to $App^{h/h}$ brains (*Figure 2D*, upper panel: quantification in *Figure 2L* -P = 0.0014). In contrast, sAPPα was significantly higher in $App^{p/p}$ as compared to $App^{h/h}$ brains (*Figure 2D*, lower panel: quantification in *Figure 2M* -P = 0.0001). These findings support the notion that the APPp mutant expressed by the $App^p$ allele is cleaved more efficiently by α-secretase and less efficiently by β-secretase, just like the human counterpart.

To validate further these findings, we quantified the other APP metabolites detected by WB analysis (APP, αCTF and βCTF, *Figure 2A and C*) as well as Aβ peptides by ELISA. APP levels are unchanged in $App^{p/p}$ brains (*Figure 2E*, APP levels detected by 6E10 -P = 0.4586-, *Figure 2F* APP levels detected by Y188 -P = 0.9385). βCTF levels detected by 6E10 were significantly lower in $App^{p/p}$ as compared to $App^{h/h}$ brains (*Figure 2G*, p=0.0484): in contrast quantification of βCTF levels detected by Y188 did not show significant differences between these two genotypes (*Figure 2H*, p=0.4109). Quantification of βCTF levels in 6E10 blots, where αCTF is not detected (*Figure 2C*), is more accurate than quantification of βCTF levels in Y188 blots (*Figure 2A*), where αCTF, which runs very close to βCTF, is the predominant APP-CTF species. Levels of αCTF, albeit slightly higher in $App^{p/p}$ rats, where also not significantly different between $App^{p/p}$ and $App^{h/h}$ brains (*Figure 2I*, p=0.3121). We also analyzed the ratio of βCTF detected either by Y188 or 6E10 to αCTF detected

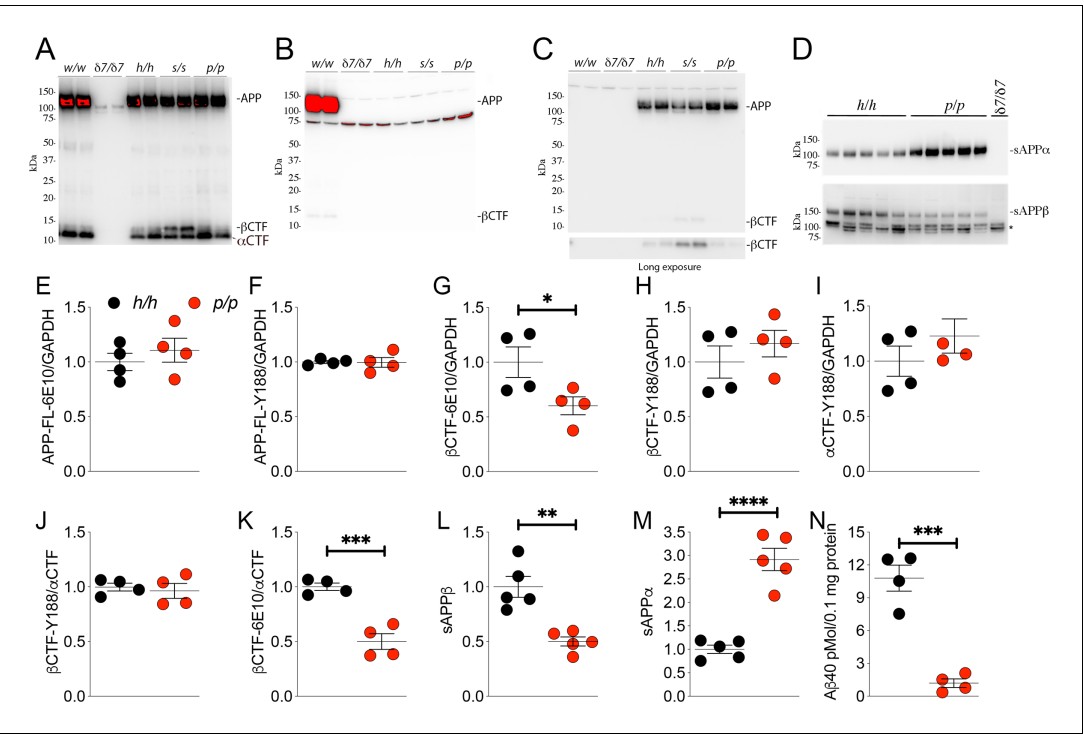

**Figure 2.** The protein encoded by the $App^p$ allele contains the humanizing and protective mutations, which reduces β-processing and increases α-processing of APP. Western blot (WB) analysis of brain lysate isolated from $App^{w/w}$, $App^{\delta7/\delta7}$, $App^{h/h}$, $App^{s/s}$, and $App^{p/p}$ rats with: (A) Y188, an antibody that detects mAPP, imAPP, αCTF, and βCTF. Specific APP signals are detected from all animals except the $App^{\delta7/\delta7}$ rats); (B) M3.2, a mouse monoclonal antibody that detects only rat WT APP and βCTF; (C) 6E10, a mouse monoclonal antibody that detects only APP and βCTF carrying the humanizing mutations. (D) WB analysis with anti-sAPPα and anti-sAPPβ-WT (absent in $App^{\delta7/\delta7}$ controls, *=non specific signal). (E–K) Quantification of APP metabolites in $App^{h/h}$ and $App^{p/p}$ rats normalized to GAPDH; APP levels as detected by either 6E10 (E) or Y188 (F); βCTF levels as detected by either 6E10 (G) or Y188 (H); αCTF levels as detected by Y188 (I); βCTF/αCTF ratio as measured by either 6E10-βCTF (J) or Y188-βCTF (K) quantitation values. Quantification of sAPPβ (L) and sAPPα (M) WB levels. (N) Aβ40 levels in $App^{h/h}$ and $App^{p/p}$ samples measured by Wako ELISA. Overexposed WBs are provided in panels A-B to show CTF levels clearly. Quantitations were performed on non-saturated exposures. Data are represented as mean ± SEM. Statistical analyses are by unpaired student's t-test (*p<0.05; **p<0.01; ***p<0.001; ****p<0.0001). Animals were analyzed at p21. We used the following male and female animals: 2E, F, G, H, I, J, K, N; $App^{h/h}$, 2 males and 2 females; $App^{p/p}$, 2 males and 2 females: 2L and 2M; $App^{h/h}$, 3 males and 2 females; $App^{p/p}$, 3 males and 2 females.

The online version of this article includes the following source data for figure 2:

**Source data 1.** Related to *Figure 2E,F,G,H,I,J,K,L,M*.

by Y188, βCTF-Y188/αCTF, and -βCTF-6E10/αCTF, respectively. While the βCTF-Y188/αCTF ratio was not different in $App^{p/p}$ and $App^{h/h}$ samples (*Figure 2J*, p=0.6681), the βCTF-6E10/αCTF ratio was significantly lower in $App^{p/p}$ brains (*Figure 2K*, p=0.0007). Finally, we measured Aβ40 peptides (human Aβ40 since our KI rats produce human Aβ species) by ELISA. As shown in *Figure 2N*, Aβ40 levels are significantly reduced in $App^{p/p}$ as compared to $App^{h/h}$ brains (p=0.0003). In summary, the data shown in *Figure 2* suggest that our $App^p$ KI rats reproduce the APP metabolic changes caused by the Icelandic protective mutation in humans: that is reduced rate of APP processing by β-secretase and increased rate of APP cleavage by α-secretase.

Next, we analyzed a new cohort of animals for the following reasons: 1) to determine reproducibility of our findings in a different rat cohort that includes a larger number of subjects; 2) to test whether these APP metabolic changes are also evident in heterozygous $App^{p/h}$ rats, which genocopy the condition that protects humans from dementia and normal cognitive decline; 3) to determine

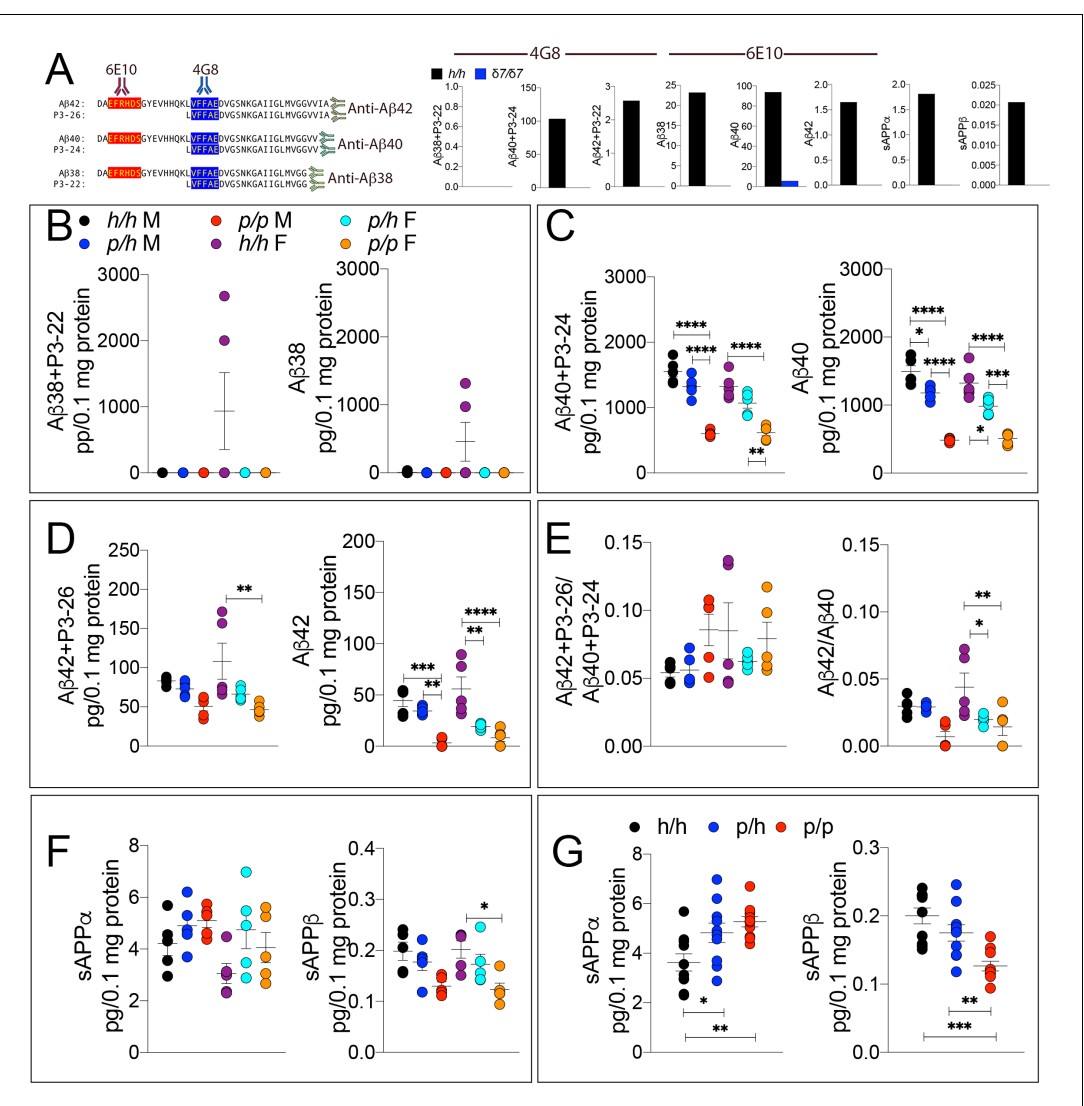

**Figure 3.** ELISA of APP metabolites in *App^p* rats shows decreased APP processing by β-secretase and increased processing by α-secretase. To test whether the APP protective mutation results in the expected changes in APP metabolism, we extracted brain tissue from a larger cohort that can identify sex-dependent changes in *App^h/h*, *App^p/h* and *App^p/p* 28 day old rats. (**A**) Left panel, Amino acid sequence of Aβ and P3 peptides recognized by Meso Scale Discovery kits. Detection antibody epitopes for 6E10 (red) and 4G8 (blue) are shown. Capture antibodies recognize Aβ42 and P3-26 (top), Aβ40 and P3-24 (middle), and Aβ38 and P3-22 (bottom). Right panel, Validation of ELISAs of Aβ/P3-4G8, Aβ−6E10, sAPPα, and sAPPβ using *App^δ7/δ7* and *App^h/h* brain lysates. (**B**) ELISA of brain lysates for Aβ38 and P3-22 (left) and Aβ38 (right) showed that no significant differences between in *App^h/h*, *App^p/h* and *App^p/p* rats. (**C**) ELISA of brain lysates for Aβ40 and P3-24 (left) and Aβ40 (right) showed a gene-dose dependent and sex independent decrease as follows: *App^h/h* > *Appp^p/h* > *Appp^p/p*. (**D**) ELISA of brain lysates for Aβ42 and P3-26 (left) and Aβ42 (right) showed a gene-dose dependent and sex independent decrease as follows: *App^h/h* > *Appp^p/h* > *Appp^p/p*. (**E**) Aβ42+P3-26/Aβ40+P3-24 ratio and Aβ42/Aβ40 ratio. (**F**) ELISA of sAPPα and sAPPβ of *App^h/h*, *App^p/h*, and *App^p/p* rat brains separated by sex and (**G**) with sexes pooled. Data are represented as mean ± SEM. Data were analyzed by Ordinary one-way ANOVA followed by post-hoc Tukey's multiple comparisons test when ANOVA showed statistically significant differences. Animals were analyzed at p28. We used 5 male and 5 female rats for each genotype. To reduce complexity of the panels, in the graphs with both sexes only intra-sex differences are shown (*p<0.05; **p<0.01; ***p<0.001; ****p<0.0001).

The online version of this article includes the following source data for figure 3:

**Source data 1.** Related to *Figure 3B,C,D,E,F,G*.

whether sex influences these alterations; 4) to measure other Aβ and P3 species; 5) to measure sAPPα and sAPPβ in total brain homogenates.

The WAKO Aβ42 ELISA kit could not reliably measure Aβ42, which is considered the main pathogenic Aβ species in humans, in $App^{p/p}$ rats (data not shown). Thus, we tested several other ELISA kits including the MSD Aβ38/Aβ40/Aβ42 ELISA kit. There are two versions of this kit: one which uses 6E10 as detection antibody and one that uses 4G8 as detection antibody. As shown in *Figure 3A*, 6E10 will only measure Aβ38, Aβ40 and Aβ42; in contrast, 4G8 will detect Aβ38, Aβ40, Aβ42 as well as the corresponding P3 peptides P3-22, P2-24 and P3-26, respectively. Thus, comparing the signal obtained with the two kits may yield information concerning levels of both Aβ and P3 peptides (*Siegel et al., 2017*). To determine specificity of these kits, we tested brain homogenates derived from $App^{δ7/δ7}$, in which the Aβ-coding region is out of frame with App due to the 7 base pair deletion and that at any rate results in a hypomorphic allele (*Tambini et al., 2019*), and $App^{h/h}$ rats. Both kits detect a specific signal -that is in $App^{h/h}$ but not $App^{δ7/δ7}$ brain homogenates (*Figure 3A*). Having established the validity of these ELISA assays, we analyzed our experimental samples. Aβ38+P3-22 (F (5, 24)=2.578, p=0.0529) and Aβ38 (F (5, 24)=2.554, p=0.0546) were detectable in a few $App^{h/h}$ brain homogenates only (*Figure 3B*). Aβ40+P3-24 (F (5, 24)=34.30, p<0.0001), Aβ40 (F (5, 24)=43.32, p<0.0001), Aβ42+P3-26 (F (5, 24)=5.062, p=0.0026) and Aβ42 (F (5, 24)=13.76, p<0.0001) peptides were significantly reduced, in a gene-dosage-dependent manner, in $App^{p/h}$ and $App^{p/p}$ rats (*Figure 3C and D*). We did not detect any sex-specific effect on Aβ/P3 peptides production. In addition, while the Aβ42+P3-26/Aβ40+P3-24 ratio was unchanged (F (5, 24)=1.694, p=0.1745), analysis of the Aβ42/Aβ40 ratio suggests that the Icelandic protective allele may reduce this ratio (*Figures 3E,F* (5 and 24)=5.756, p=0.013), which could underlie a protective mechanism prompted by the Icelandic variant. However, more animals need to be analyzed to determine the significance of this observation. As for P3 levels, we rejected the idea of measuring P3 peptides by subtracting the 6E10 measurements from the 4G8 measurements because in several brain samples the 6E10 signal was higher than the 4G8 signal. This suggests that the two kits have different efficiencies. Although the significance of the differences among genotypes is higher in 6E10 measurements as compared to 4G8 measurements -this is obviously evident when comparing Aβ42+P3-26 and Aβ42 ELISAs- suggesting that P3 peptides may be increased in $App^{p/h}$ and $App^{p/p}$ rats, we cannot draw definitive conclusions concerning levels of P3 and whether these levels are changed in a genotype-dependent manner. Nevertheless, the finding that Aβ peptides are consistently reduced using two different detection antibodies stresses the significance of the finding. In addition, they exclude the possibility that the reduction of Aβ peptides in $App^p$ rats seen in the 6E10-based ELISA may be due to a reduced affinity of Aβ species carrying the protective Icelandic mutation, which is adjacent to the 6E10 epitope (*Figure 1A*), for 6E10.

Next, we measure sAPPα and sAPPβ using the MSD ELISA kit, which is also specific (see *Figure 3A*, the two panels in the right end-side showing signals in $App^{h/h}$ but not $App^{δ7/δ7}$ rats), in total brain homogenates rather than on soluble brain fractions, as done for the experiments shown in *Figure 2D,L and M*. We used total brain homogenates for the following reasons. While α-secretase cleaves APP in the secretory pathway or on the plasma membrane releasing sAPPα into the extracellular fluid, β-secretase cleaves mAPP in acidic organelles, including synaptic vesicles and late endosomes. A fraction of sAPPβ is released extracellularly during exocytosis but significant amounts of sAPPβ are present in intracellular compartments. Analysis of ELISA by sex showed some significant decrease in sAPPβ (F (5, 24)=4.552, p=0.0046), but not in sAPPα (F (5, 24)=2.263, p=0.0806) (*Figure 3F*). When data from males and females were pooled, we detected a gene-dosage-dependent increase in sAPPα (F (5, 24)=6.725, p=0.0043) and a significant decrease in sAPPβ (F (5, 24)=12.64, p=0.0001) in $App^{p/p}$ rats (*Figure 3G*).

Samples from these 30 animals were then analyzed by WB. Examples of the WB analyses with Y188 and an anti-GAPDH antibody are shown in *Figure 4H,I and J*. Total APP levels were not changed in either male or females (*Figure 5A*, males F (2, 12)=1.756, p=0.2144; females (F (2, 12)=2.367, p=0.1360). Quantification of mature APP (mAPP), immature APP (imAPP), αCTF and βCTF showed the following: levels of mAPP (males F (2, 12)=1.493, p=0.2635; females F (2, 12)=2.090, p=0.1615, *Figure 4C*) and imAPP were not significantly changed -except for a reduction in imAPP levels in female $App^{p/h}$ brains (males F (2, 12)=1.036, p=0.3846; females F (2, 12)=4.180, p=0.0419, *Figure 4B*), which may or may not be confirmed in further analyses- in animals carrying the protective Icelandic mutation. The mAPP/imAPP ratio was also not changed (*Figure 4D*, males F (2, 12)

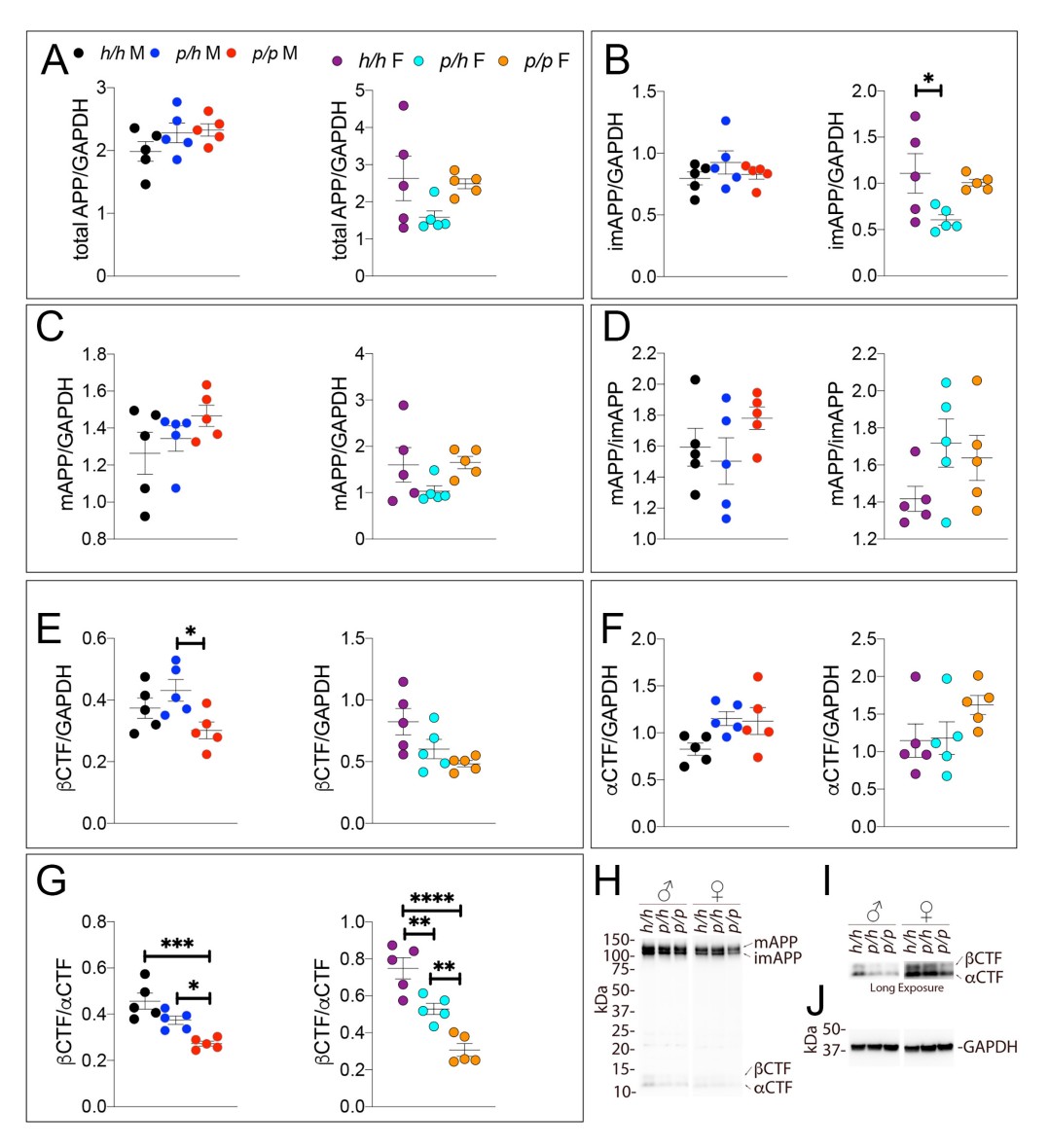

**Figure 4.** Western analysis of APP metabolites in $App^p$ rats shows decreased APP processing by β-secretase. Quantitation of WB analysis (**A–G**) with representative blots (**H–J**). (**A**) Normalized total APP levels in $App^{h/h}$, $App^{p/h}$ and $App^{p/p}$ male (left) and female (right) rats. (**B**) Normalized imAPP levels in $App^{h/h}$, $App^{p/h}$ and $App^{p/p}$ male (left) and female (right) rats. (**C**) Normalized mAPP levels in $App^{h/h}$, $App^{p/h}$ and $App^{p/p}$ male (left) and female (right) rats. (**D**) Ratio of mAPP:imAPP in $App^{h/h}$, $App^{p/h}$ and $App^{p/p}$ rats. (**E**) Normalized β-CTF levels in $App^{h/h}$, $App^{p/h}$ and $App^{p/p}$ male (left) and female (right) rats. (**F**) Normalized α-CTF levels in $App^{h/h}$, $App^{p/h}$ and $App^{p/p}$ male (left) and female (right) rats. (**G**) Ratio of β-CTF: α-CTF. (**H**) Representative blot against C-terminus of APP (**I**) Longer exposure of α-APP-C-terminus blot to detect α-CTF and β-CTFs. (**J**) Anti-GAPDH loading control. Data are represented as mean ± SEM. Data were analyzed by Ordinary one-way ANOVA followed by post-hoc Tukey's multiple comparisons test when ANOVA showed statistically significant differences (*p<0.05; **p<0.01; ***p<0.001; ****p<0.0001). Animals were analyzed at p28. We used 5 male and 5 female rats for each genotype. The online version of this article includes the following source data and figure supplement(s) for figure 4:

**Source data 1.** Related to *Figure 4A,B,C,D,E,F,G*.
**Figure supplement 1.** Western blots images used for quantitation shown in *Figure 4*.

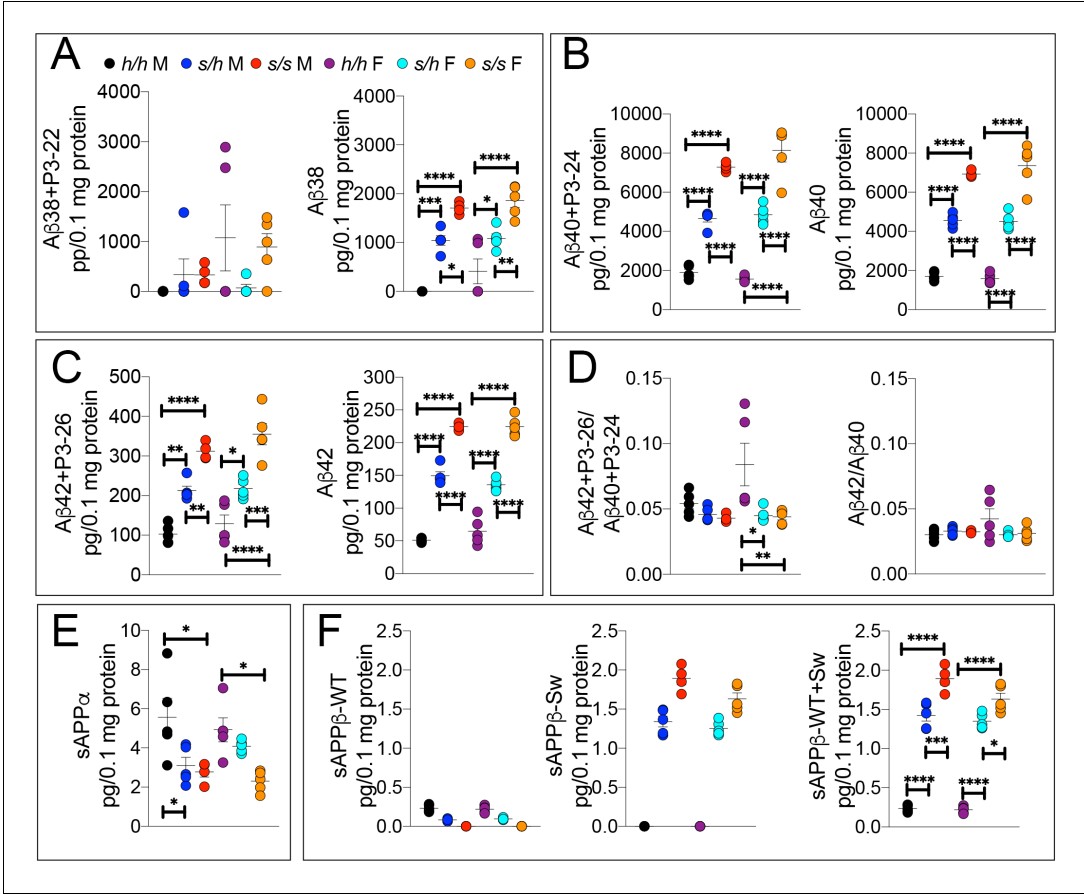

**Figure 5.** ELISA of APP metabolites in *App*<sup>s</sup> rats shows increased APP processing by β-secretase and decreased processing by α-secretase. To confirm the Swedish mutation results in the expected changes in APP metabolism, we extracted brain tissue from a larger cohort that can identify sex-dependent changes in $App^{h/h}$, $App^{s/h}$, and $App^{s/s}$ 28 day old rats. (**A**) ELISA of brain lysates for Aβ38 and P3-22 (left) showed that no significant differences between in $App^{h/h}$, $App^{s/h}$, and $App^{s/s}$ when P3-22 is considered, however Aβ38 (right) showed a gene-dose dependent and sex independent increase as follows: $App^{h/h} < Apps^{s/h} < Apps^{s/s}$. (**B**) ELISA of brain lysates for Aβ40 and P3-24 (left) and Aβ40 (right) showed a gene-dose dependent and sex independent decrease as follows: $App^{h/h} < Apps^{s/h} < Apps^{s/s}$. (**C**) ELISA of brain lysates for Aβ42 and P3-26 (left) and Aβ42 (right) showed a gene-dose-dependent and sex-independent decrease as follows: $App^{h/h} < Apps^{s/h} < Apps^{s/s}$. (**D**) Aβ42+P3-26/Aβ40 +P3-24 ratio and Aβ42/Aβ40 ratio. (**E**) ELISA of sAPPα (**F**) ELISA of sAPPβ-WT (left) sAPPβ-Sw (middle) and the calculated sum of both sAPPβ species (right). Data are represented as mean ± SEM. Data were analyzed by Ordinary one-way ANOVA followed by post-hoc Tukey's multiple comparisons test when ANOVA showed statistically significant differences. Animals were analyzed at p28. We used 5 male and 5 female rats for each genotype, except for male $App^{s/s}$ (n = 4). To reduce complexity of the panels, in the graphs with both sexes only intra-sex differences are shown (*p<0.05; **p<0.01; ***p<0.001; ****p<0.0001).

The online version of this article includes the following source data for figure 5:

**Source data 1.** Related to *Figure 5A,B,C,D,E,F*.

=1.406, p=0.2828; females F (2, 12)=2.009, p=0.1768). βCTF levels were significantly lower in $App^{p/p}$ females as compared to $App^{h/h}$ females (*Figure 4E*, males F (2, 12)=4.131, p=0.0431; females F (2, 12)=4.873, p=0.0282). Levels of αCTF, albeit slightly higher in $App^{p/p}$ rats, were also not significantly different (*Figure 4F*, males F (2, 12)=3.276, p=0.0732; females F (2, 12)=1.869, p=0.1965). However, the βCTF/αCTF ratio was decreased significantly in a gene-dosage dependent manner in both sexes (*Figure 4G*, males F (2, 12)=15.15, p=0.0005; females F (2, 12)=27.27, p<0.0001). The blots used for

quantitation are shown in *Figure 4—figure supplement 1*. Overall, the data suggest that the protective Icelandic mutation reduced the rate of APP processing by β-secretase and increased rate of APP cleavage by α-secretase in a gene-dosage dependent manner.

## Gene-dosage-dependent decreased α-processing and increased β-cleavage of APP in *App^s* Knock In rats

We have recently shown that *App^{s/s}* rats, that is rats carrying two *App* alleles with the humanizing and pathogenic Swedish mutations, show an opposite phenotype: that is reduced the rate of APP processing by α-secretase and increased rate of APP cleavage by β-secretase (*Tambini et al., 2019*). To confirm these findings, to test whether these APP metabolic changes are also evident in heterozygous *App^{s/h}* rats, which genocopy the condition that causes dementia in humans, and to determine whether sex influences these alterations, we analyzed a new cohort of *App^{h/h}*, *App^{s/h}* and *App^{s/s}* rats.

While Aβ38+P3-22 levels were not significantly altered by either sex or genotype (*Figures 5A,F* (5 and 23)=1.779, p=0.1570), Aβ38 (*Figures 5A,F* (5 and 23)=27.42, p<0.0001), Aβ40+P3-24 (*Figures 5B,F* (5 and 23)=86.85, p<0.0001), Aβ40 *Figures 5B,F* (5 and 23)=104.07, p<0.0001), Aβ42 +P3-26 (*Figures 5C,F* (5 and 23)=32.72, p<0.0001) and Aβ42 (*Figures 5C,F* (5 and 23)=166.2, p<0.0001) peptides were significantly increased in a sex-independent but gene-dosage-dependent manner in *App^{s/h}* and *App^{s/s}* rats as compared to *App^{h/h}* animals. We did not detect any sex specific effect on Aβ/P3 peptides production. The Aβ42+P3-26/Aβ40+P3-24 ratio was highest in *App^{h/h}* female rats (*Figures 5D,F* (5 and 23)=4.745, p=0.0040) while the Aβ42/Aβ40 ratio was similar in all rats (*Figures 5D,F* (5 and 23)=1.735, p=1667). Further experiments will be needed to verify the reproducibility of this finding. Even for this set of animals, the significance of the differences among genotypes is higher in 6E10 measurements as compared to 4G8 measurements, once again especially when comparing Aβ42+P3-26 and Aβ42 ELISAs- suggesting that P3 peptides may be decreased *App^{s/h}* and *App^{s/s}* rats.

Next, we measured sAPPα and sAPPβ. As shown in *Figure 5E* (F (5, 23)=5.762, p=0.0014), sAPPα is significantly reduced in both *App^{s/h}* and *App^{s/s}* male rats. Female rats show a significant sAPPα steady-state levels reduction only in homozygous *App^{s/s}* rats. The sAPPβ produced by β-cleavage of the Swedish APP mutant -which is called sAPPβ-Sw- differs at the COOH-terminus from the sAPPβ produced by β-cleavage of humanized APP mutant -which we refer to as sAPPβ-WT- since the KM > NL mutations are in the 2 COOH-terminal residues of sAPPβ moieties. Thus, the MSD sAPPα/ sAPPβ MSD ELISA kit will only recognize sAPPβ-WT, as shown by the fact that this kit does not detect sAPPβ in *App^{s/s}* rats (*Figure 5F*, left panel). To measure sAPPβ-Sw we used the MSD ELISA kit K151BUE that, conversely, detects sAPPβ-Sw molecules in *App^{s/h}* and *App^{s/s}* but not *App^{h/h}* rats (*Figure 5F*, middle panel). The first striking observation is that, in *App^{s/h}* rats, levels of sAPPβ-Sw are ~10 times higher than the levels of sAPPβ-WT suggesting that in these animals, APPSw is cleaved by β-secretase at a rate 10 times higher than APPwt. Even if these two kits have slightly different efficiencies, as observed earlier for the 6E10 and 4G8 ELISA kits, it is highly improbable that they may account for this result. To compare the total sAPPβ steady-state levels in these rats, we added the signals detected for sAPPβ-WT to those detected for sAPPβ-Sw (*Figure 5F*, left panel). Total sAPPβ is significantly increased in a gene-dosage dependent manner in both sexes: in addition, male *App^{s/s}* rats had significantly higher sAPPβ-Sw levels as compared to female *App^{s/s}* rats (F (5, 23)=163.1, p<0.0001).

Western analysis was performed on these 30 samples to detect changes in APP-CTF levels consistent with a shift toward β-processing of APP. Examples of the WB analyses with Y188 and an anti-GAPDH antibody are shown in *Figure 6H,I and J*. Probing against the C-terminus of APP revealed a trend towards decreased amounts of total full-length APP in *App^{s/s}* rats, which reached significance only in male *App^{s/s}* rats compared to *App^{h/h}* rats (*Figure 6A*, males F (2, 11)=11.08, p=0.0023; females F (2, 12)=2.987, p=0.0886). Separating out the glycosylated, mature form, there was no statistical difference in imAPP levels in any of the genotypes of each sex (*Figure 6B*, males F (2, 11) =2.125, p=0.1658; females F (2, 12)=1.516, p=0.2589), however, a significant decrease was seen in mAPP levels in *App^{s/s}* rats compared to *App^{h/h}* (*Figure 6C*, males F (2, 11)=15.57, p=0.0006; females F (2, 12)=5.053, p=0.0256). The data suggest a trend toward a gene-dose dependent decrease in mAPP levels, but the data are not significant for all comparisons and, when mAPP levels are normalized to imAPP levels, the effect is only seen in *App^{s/s}* rats, with no sex differences

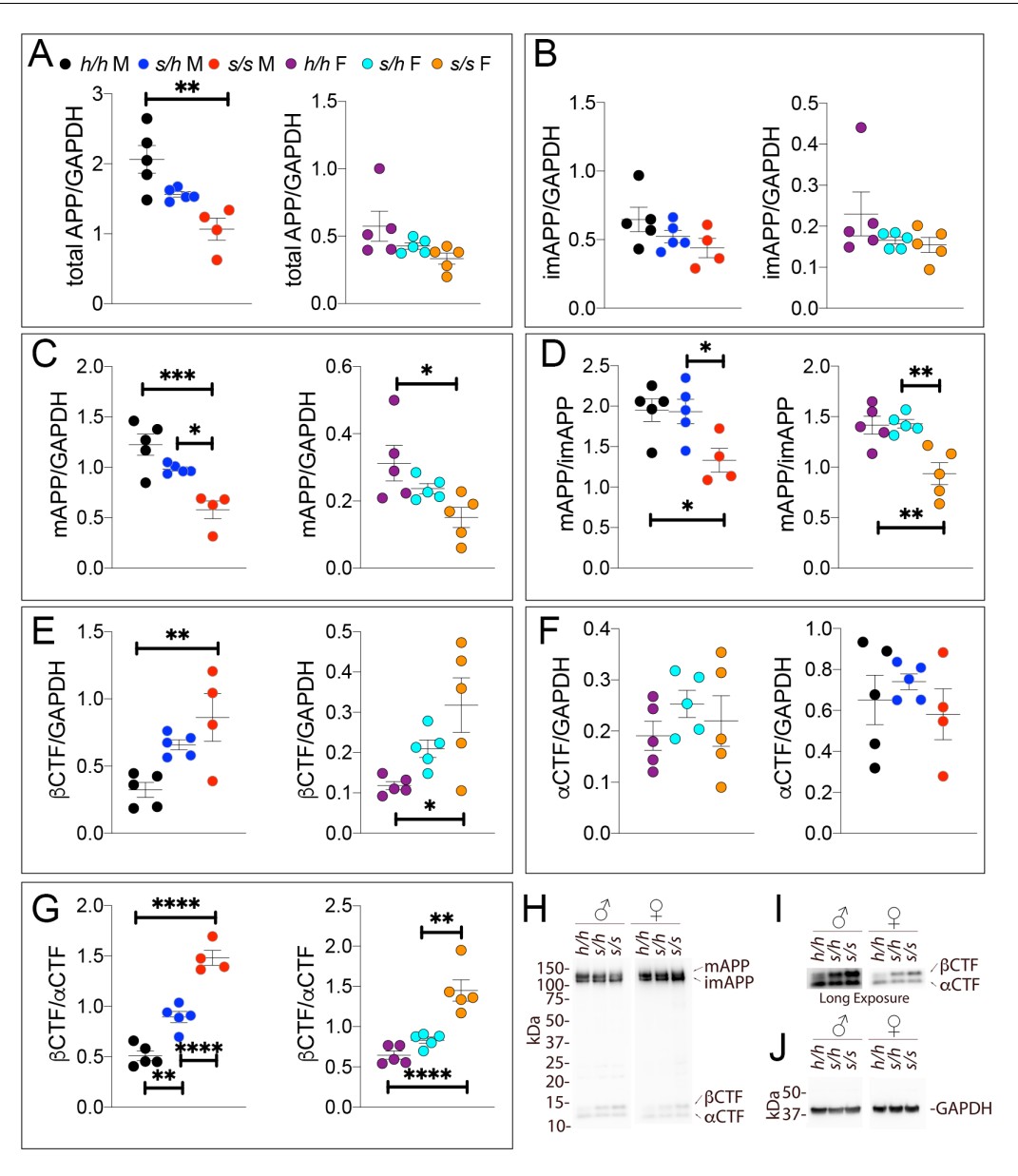

**Figure 6.** Western analysis of APP metabolites in $App^s$ rats shows increased APP processing by β-secretase and decreased levels of mature APP. Quantitation of WB analysis (**A–G**) with representative blots (**H–J**). (**A**) Normalized total APP levels in $App^{h/h}$, $App^{s/h}$, and $App^{s/s}$ male (left) and female (right) rats, show a gene-dose dependent decrease in $App^s$ rats. (**B**) Normalized imAPP levels in $App^{h/h}$, $App^{s/h}$, and $App^{s/s}$ male (left) and female (right) rats. (**C**) Normalized mAPP levels in $App^{h/h}$, $App^{s/h}$, and $App^{s/s}$ male (left) and female (right) rats show a gene-dose dependent decrease in $App^s$ rats. (**D**) Ratio of mAPP:imAPP in $App^{h/h}$, $App^{s/h}$, and $App^{s/s}$ rats. (**E**) Normalized βCTF levels in $App^{h/h}$, $App^{s/h}$, and $App^{s/s}$ male (left) and female (right) rats. (**F**) Normalized αCTF levels in $App^{h/h}$, $App^{s/h}$, and $App^{s/s}$ male (left) and female (right) rats. (**G**) Ratio of βCTF: αCTF. (**H**) Representative blot against C-terminus of APP (**I**) Longer exposure of α-APP-C-terminus blot to detect αCTF and βCTFs. (**J**) Anti-GAPDH loading control. Data are represented as mean ± SEM. Data were analyzed by Ordinary one-way ANOVA followed by post-hoc Tukey's multiple comparisons test when ANOVA showed statistically significant differences (*p<0.05; **p<0.01; ***p<0.001; ****p<0.0001.. Animals were analyzed at p28. We used 5 male and 5 female rats for each genotype, except for male $App^{s/s}$ (n = 4).

The online version of this article includes the following source data and figure supplement(s) for figure 6:

**Source data 1.** Related to *Figure 6A,B,C,D,E,F,G*.

**Figure supplement 1.** Western blots images used for quantitation shown in *Figure 6*.

(*Figure 6D*, males F (2, 11)=5.292, p=0.0245; females F (2, 12)=11.05, p=0.0019). This decrease, which we also observed in a different rat cohort (*Tambini et al., 2019*), is not likely the result of decreased expression of APP, given that APP mRNA levels are unchanged in this model (*Tambini et al., 2019*), but may reflect, given the large increase in Aβ and sAPPβ-Sw production (*Figure 5*), a β-processing driven depletion of the total pool of mAPP. Previous generation of a similar *App*-KI mouse model of the Swedish mutation did not reveal any alteration in mAPP levels, though this possibility was not specifically tested (*Saito et al., 2014*).

βCTF levels were significantly increased in *App^{s/s}* rats of both sexes (*Figure 6E*, males F (2, 11)=7.926, p=0.0074; females F (2, 12)=5.819, p=0.0171). Levels of αCTF were not significantly different (*Figure 6F*, males F (2, 11)=0.620, p=0.5557; females (F (2, 12)=0.732, p=0.5012). The βCTF/αCTF ratio was increased significantly in a gene-dosage dependent manner in both sexes (*Figure 6G*, males F (2, 11)=66.89, p<0.0001; females F (2, 12)=24.78, p<0.0001). The blots used for quantitation are shown in *Figure 6—figure supplement 1*.Overall, the data suggest that this pathogenic mutation reduced the rate of APP processing by α-secretase and increased rate of APP cleavage by β-secretase in a gene-dosage dependent manner.

## Opposite and gene-dosage-dependent changes in βCTF steady-state levels in *App^P* and *App^s* Knock In rats

As stated above, 6E10, which detects βCTF but not αCTF, can more accurately measure βCTF levels. Thus, we performed 6E10 WB analysis of Swedish and Icelandic KI rat brain lysates. As shown in *Figure 7A and B*, *App* Swedish rats show a significant decrease in APP (females F (2, 12)=67.62, p<0.0001; males F (2, 11)=5.760, p=0.0194) and a significant increase in βCTF steady-state levels (females F (2, 12)=49.69, p<0.0001; males (F (2, 11)=28.55, p<0.0001): these changes are gene-dosage dependent. The blots used for quantitation are shown in *Figure 7—figure supplement 1A and B*.

Given the more subtle changes in APP processing caused by the Icelandic mutation, we compared control *App^{h/h}* animals to homozygous *App^{p/p}* KI rats. Assessments of βCTF and APP levels (representative Western blots are shown in *Figure 7C*) showed a decrease in βCTF (*Figure 7D*); this decrease was significant in females (p=0.0023), trended toward significance in males (p=0.0549), and significance increased when male and female data were grouped together (p=0.0007). In contrast, APP levels were similar in *App^{h/h}* and *App^{p/p}* animals (*Figure 7E*, females p=0.2123; males p=0.4846; females + males p=0.1810). To analyze further βCTF levels, we used 3 internal controls and expressed βCTF as ratios of these controls: 1) βCTF/APP, which measures the relative abundance of βCTF compared to its precursor APP (*Figure 7F*); 2) βCTF/GAPDH, which measures the relative abundance of βCTF to the housekeeping protein GAPDH (*Figure 7G*, this control is widely used based on the assumption, which may not be always correct, that GAPDH levels remain always constant); 3) βCTF/Ponc (*Figure 7H*), which measures the relative abundance of βCTF compared to the total amount of proteins blotted on the Western blot membrane as determined by ponceau stain and is, in theory, a better control. All 3 analyses indicate that βCTF levels are significantly reduced in female *App^{p/p}* rats (*Figure 7F*, p<0.0001; *Figure 7G*, p=0.0002; *Figure 7H*, p=0.0083). As for the males, the reduction in βCTF in *App^{p/p}* rats was significant for the βCTF/APP and βCTF/GAPDH ratios and trended toward significance for the βCTF/Ponc ratio (*Figure 7F*, p=0.0257; *Figure 7G*, p=0.0059; *Figure 7H*, p=0.0877). The differences in βCTF levels were more obvious in female rats. Grouping male and female data increased significance (*Figure 7F*, p<0.0001; *Figure 7G*, p<0.0001; *Figure 7H*, p=0.0020). The blots used for quantitation are shown in *Figure 7—figure supplement 1C and D*.

## Swedish and protective Icelandic mutations show an additive effect and no allelic interaction on APP-processing in *App^{s/P}* Knock In rats

We wished to determine if the effects of Swedish and protective APP mutations on APP processing operate independently, or if there is interaction between the two alleles. Biochemical studies have identified that the Swedish and protective mutations result in increased and decreased affinity, respectively, of APP for BACE1. It has also been reported that APP and βCTF form homotypic dimers (*Scheuermann et al., 2001*; *Winkler et al., 2015*), therefore it is conceivable that APPSw and APPp might interact in such a way that the effect of each mutation may be abrogated or

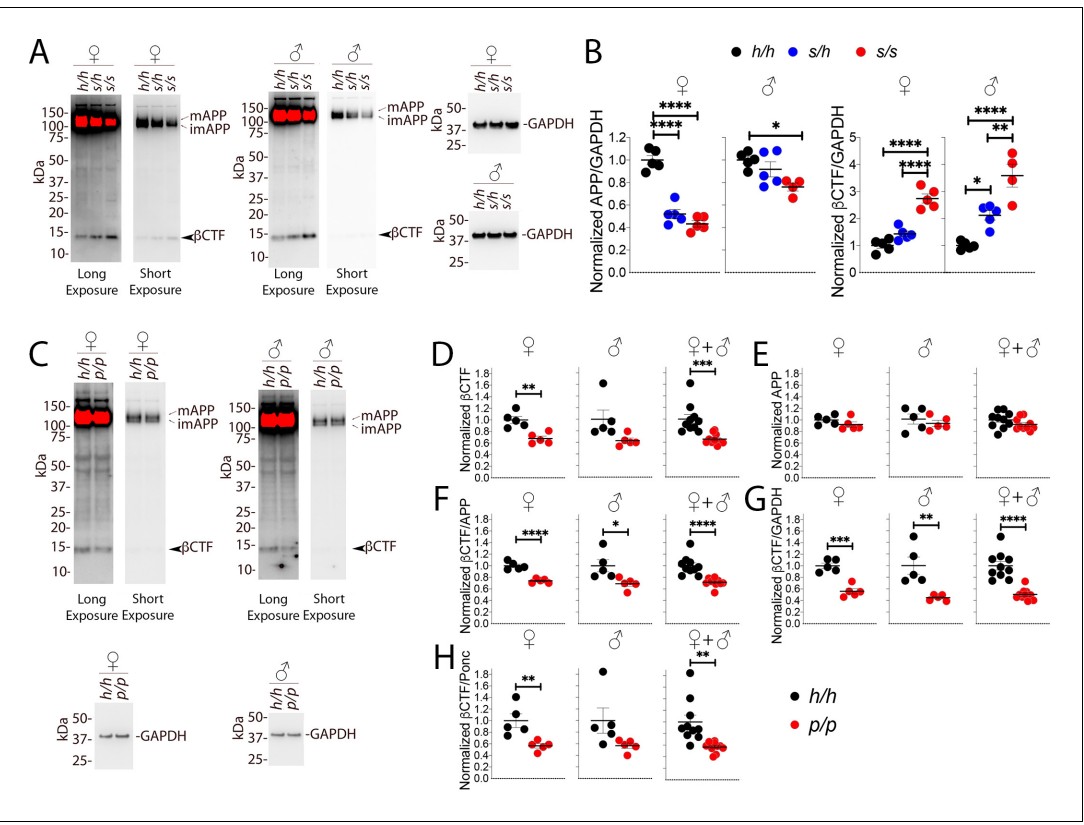

**Figure 7.** Western analysis of βCTF shows increased levels of βCTF in $App^s$ animals and decreased levels in $App^p$ rats. (**A**) Representative blot with 6E10 (Long exposure was used to quantify βCTF, Short exposure was used to quantify APP) and anti-GAPDH on $App^{h/h}$, $App^{s/h}$, and $App^{s/s}$ female and male brain lysates. The blots used for quantitation are shown in **Figure 7—figure supplement 1A and B**. (**B**) Total APP, βCTF and GAPDH were normalized against the average signal of the control $App^{h/h}$ samples and shown as ratio of the normalized APP/GAPDH and βCTF/GAPDH ratios. APP and βCTF are decrease and increased in $App^s$ rats, respectively, in a gene-dose-dependent manner. Data are represented as mean ± SEM. Data were analyzed by ordinary one-way ANOVA followed by post-hoc Tukey's multiple comparisons test when ANOVA showed statistically significant differences. Animals were analyzed at p28. We used 5 male and 5 female rats for each genotype, except for male $App^{s/s}$ (n = 4). (**C**) Representative blot with 6E10 (Long exposure was used to quantify βCTF, Short exposure was used to quantify APP) and anti-GAPDH on $App^{h/h}$ and $App^{p/p}$ female and male brain lysates. The blots used for quantitation are shown in **Figure 7—figure supplement 1C and D**. Total APP, βCTF and GAPDH were normalized against the average signal of the control $App^{h/h}$ samples. Normalized βCTF (**D**) and APP (**E**) values for female, male and female + male samples. (**F**) Normalized βCTF/APP ratios for female, male and female + male samples. (**G**) Normalized βCTF/GAPDH ratios for female, male and female + male samples. (**H**) Normalized βCTF/Ponc (Ponceau stain) ratios for female, male and female + male samples. Overall the data indicate a reduction in βCTF in $App^{p/p}$ rats. Data are represented as mean ± SEM. Statistical analyses are by unpaired student's t-test (*p<0.05; **p<0.01; ***p<0.001; ****p<0.0001). Animals were analyzed at p28. We used 5 male and 5 female rats for each genotype.

The online version of this article includes the following source data and figure supplement(s) for figure 7:

**Source data 1.** Related to **Figure 7B,C,D,E,F,G,H**.

**Figure supplement 1.** Western blots, ponceau stain and colorimetric images used for quantitation shown in **Figure 7** for: (**A**) βCTF, APP and GAPDH in $App^{h/h}$, $App^{s/h}$ and $App^{s/s}$ females; (**B**) βCTF, APP and GAPDH in $App^{h/h}$, $App^{s/h}$ and $App^{s/s}$ males (as it is evident from the blots, one $App^{s/s}$ sample was degraded and indicated with * and therefore that animal was excluded from any analysis); (**C**) βCTF, APP and GAPDH in $App^{h/h}$ and $App^{p/p}$ females; (**B**) βCTF, APP and GAPDH in $App^{h/h}$ and $App^{p/p}$ males.

partially abrogated by the other mutant allele. Alternatively, the two alleles may act independently, in which case the effect of each variant on APP processing may be additive in App$^{s/p}$ rats.

To test this possibility, we analyzed brain APP metabolites from App$^{h/h}$, App$^{s/h}$, App$^{p/h}$, and App$^{s/p}$ rats by ELISA. While Aβ38+P3-22 levels were not significantly altered by either gender sex or genotype (*Figures 8A,F* (7 and 32)=0.8586, p=0.5488), Aβ38 (*Figures 8A,F* (7 and 32)=7.310, p<0.0001), Aβ40+P3-24 (*Figures 8B,F* (7 and 32)=107.2, p<0.0001), Aβ40 (*Figures 8B,F* (7 and 32) =92.87, p<0.0001), Aβ42+P3-26 (*Figures 8C,F* (7 and 32)=42.82, p<0.0001) and Aβ42 (*Figures 8C,F* (7 and 32)=57.57, p<0.0001) peptides were significantly increased in a sex-independent but Swedish allele dosage-dependent manner in App$^{s/h}$ and App$^{s/p}$ rats as compared to App$^{h/h}$ and App$^{p/h}$ rats (*Figure 8A–C*). In general, there is a trend toward a decrease in amyloid production in App$^{s/p}$ compared to App$^{s/h}$ rats, however, the magnitude of the decrease is on the same order of magnitude as the decrease in amyloid production between App$^{h/h}$ and App$^{p/h}$ rats, which would argue against allelic interaction. The Aβ42+P3-26/Aβ40+P24 ratio was highest in App$^{p/h}$ female rats and male App$^{s/h}$ rats (*Figures 8D,F* (7 and 32)=6.875, p<0.0001), though when P3 is not considered, the Aβ42/Aβ40 ratio similar in all rats (*Figures 8D,F* (7 and 32)=2.152, p=0.0661).

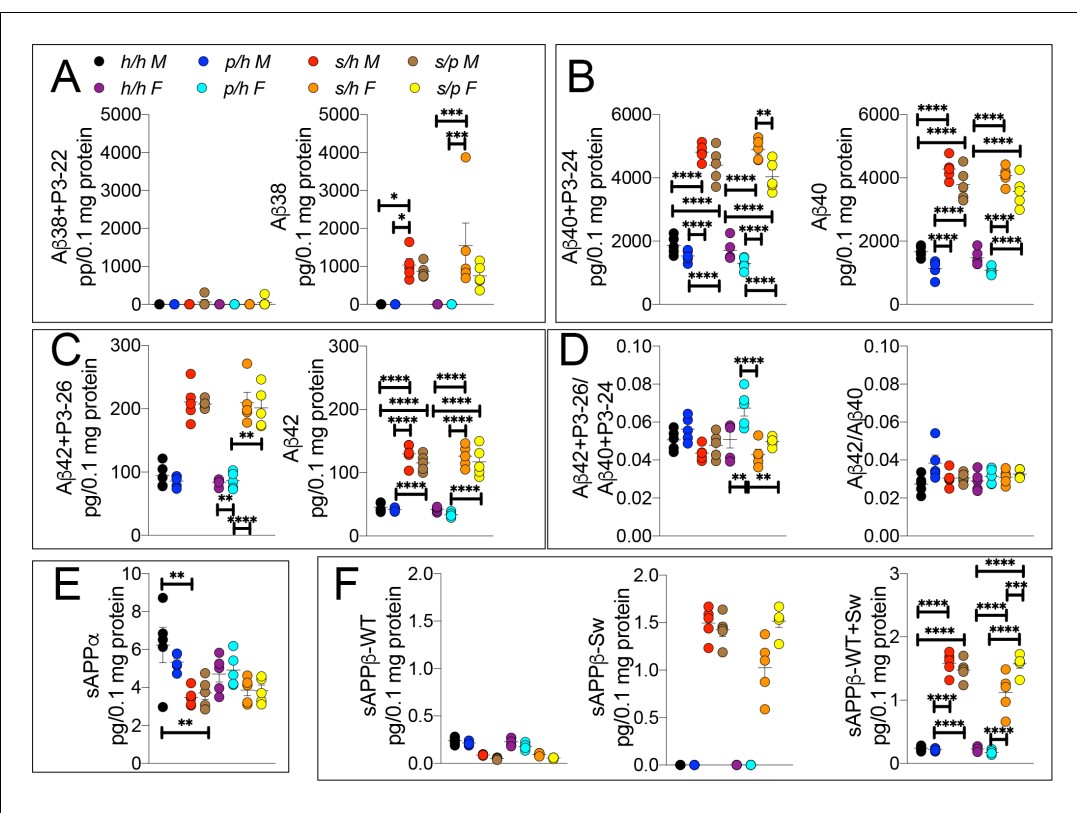

**Figure 8.** ELISA of APP metabolites in App$^{h/h}$, App$^{s/h}$, App$^{p/h}$, and App$^{s/p}$ rats. (**A**) ELISA of brain lysates for Aβ38 and P3-22 (left) and Aβ38 (right). (**B**) ELISA of brain lysates for Aβ40 and P3-24 (left) and Aβ40 (right). (**C**) ELISA of brain lysates for Aβ42 and P3-26 (left) and Aβ42 (right). (**D**) Aβ42+P3-26/Aβ40+P3-24 ratio and Aβ42/Aβ40 ratio. (**E**) ELISA of sAPPα (**F**) ELISA of sAPPβ-WT (left) sAPPβ-Sw (middle) and the calculated sum of both sAPPβ species (right). Data are represented as mean ± SEM. Data were analyzed by Ordinary one-way ANOVA followed by post-hoc Tukey's multiple comparisons test when ANOVA showed statistically significant differences. Animals were analyzed at p28. We used 5 male and 5 female rats for each genotype. To reduce complexity of the panels, in the graphs with both sexes only intra-sex differences are shown (*p<0.05; **p<0.01; ***p<0.001; ****p<0.0001). The online version of this article includes the following source data for figure 8:

**Source data 1.** Related to *Figure 8A,B,C,D,E,F*.

Levels of sAPPα and sAPPβ in $App^{h/h}$, $App^{s/h}$, $App^{p/h}$, and $App^{s/p}$ rats were analyzed by ELISA. In opposition to the data presented in *Figure 3E*, in this set of samples the protective mutation in heterozygosity does not lead to a significant increase in sAPPα levels (*Figure 8E*). The presence of a Swedish allele resulted in a decrease in sAPPα, significantly in males and only as a trend in females (*Figures 8E,F* (7 and 32)=4.657, p<0.0011), and a significant increase in sAPPβ-Sw (*Figures 8F,F* (7 and 32)=163.1, p<0.0001) for both sexes. Unexpectedly, female rats show a significant increase in sAPPβ-Sw in $App^{s/p}$ compared to $App^{s/h}$, which might suggest some degree of allele interaction, though it is unclear how valid this finding is as this decrease is not reflected in Aβ levels, where the opposite trend is seen. Overall, the data argue that the effect of the Swedish and protective mutations on APP-processing occur independently, and the magnitude of the increase in β-processing of APP caused by the Swedish mutation is not abrogated by the presence of a protective allele.

To further validate this finding, we compared the $App^{s/p}$ ELISA data to a simulated $App^{s/p}$ result calculated from ELISA data from $App^{h/h}$, $App^{s/h}$, and $App^{p/h}$ rats. The 'hypothetical' $App^{s/p}$ ELISA

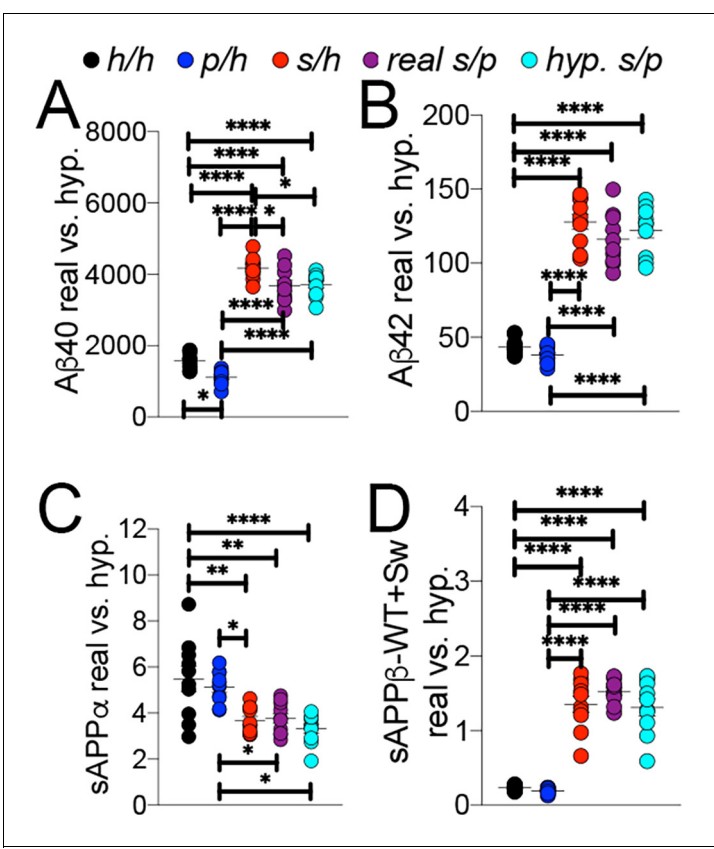

**Figure 9.** Comparison of empirical levels of APP metabolites in $App^{h/h}$, $App^{s/h}$, $App^{p/h}$, and $App^{s/p}$ rats to calculated $App^{s/p}$ levels. ELISA values from $App^{h/h}$, $App^{s/h}$, and $App^{p/h}$ rats in *Figure 7* were used to generate a hypothetical value for $App^{s/p}$ rats. One *h* allele value was calculated as half the value of $App^{h/h}$. This *h* allele value was subtracted from $App^{s/h}$ and $App^{p/h}$ values to generate the *s* and *p* allele values, respectively, which were then added together to represent a 'hypothetical' $App^{s/p}$ rat. (A) Real and calculated ELISA Aβ40 levels. (B) Real and calculated ELISA Aβ42 levels. (C) Real and calculated ELISA sAPPα levels. (D) Real and calculated ELISA total sAPPβ levels. Data are represented as mean ± SEM. Data were analyzed by Ordinary one-way ANOVA followed by post-hoc Tukey's multiple comparisons test when ANOVA showed statistically significant differences (*p<0.05; **p<0.01; ***p<0.001; ****p<0.0001). Animals were analyzed at p28. We used 5 male and 5 female rats for each genotype.

The online version of this article includes the following source data for figure 9:

**Source data 1.** Related to *Figure 9A,B,C,D*.

values were calculated as follows: one $h$ allele value was calculated as half the value of $App^{h/h}$. This $h$ allele value was subtracted from $App^{s/h}$ and $App^{p/h}$ to generate the $s$ and $p$ allele values, respectively, which were then added together to represent a 'hypothetical' $App^{s/p}$ rat. These calculations (*Figure 9*) were performed using the $App^{s/p}$ ELISA results in *Figure 8B,C,E and F*, right panels, with male and female data pooled by genotype. There is no significant difference with regard to levels of Aβ40 (*Figures 9A,F* (4 and 45)=187.6, p<0.0001), Aβ42 (*Figures 9B,F* (4 and 45)=113.3, p<0.0001), sAPPα (*Figures 9C,F* (4 and 45)=9.893, p<0.0001) and sAPPβ (*Figures 9D,F* (4 and 45)=76.86, p<0.0001) between real and hypothetical $App^{s/p}$ rats, which is consistent with the idea that each allele's effect on APP metabolism is independent and additive.

We next performed Western analysis of $App^{h/h}$, $App^{s/h}$, $App^{p/h}$, and $App^{s/p}$ rat brains to measure levels of APP and its metabolites. Examples of the WB analyses with Y188 and an anti-GAPDH antibody are shown in *Figure 10H–J*. Total APP levels (*Figure 10A*, males F (3, 16)=1.725, p=0.2022; females F (3, 16)=6.525, p=0.0043) and mAPP (*Figure 10C*, males F (3, 16)=2.794, p=0.0739; females F (3, 16)=7.551, p=0.0023), but not imAPP (*Figure 10B*, males F (3, 16)=0.5694 P=0.6432; females F (3, 16)=3.194, p=0.0520), are significantly lower in female $App^{s/p}$ rats. Although βCTF (*Figure 10E*, males F (3, 16)=1.930, p=0.1654; females F (3, 16)=1.903, p=0.1699) and αCTF (*Figure 10F*, males F (3, 16)=1.481, p=0.2573; females F (3, 16)=2.582, p=0.0896) were not significantly different, we observed an expected increase in the βCTF/αCTF ratio for all animals carrying a Swedish allele (*Figure 10G*, males F (3, 16)=13.04, p<0.0001; females F (3, 16)=18.58, p<0.0001). The blots used for quantitation are shown in *Figure 10—figure supplement 1*.

## Discussion

The finding that the A673T *APP* mutation protects against late-onset AD and non-AD dementia, is notable for several reasons. First, it links APP processing to the sporadic forms of dementia. Previously, though amyloid plaques linked the histopathology of early and late-onset AD (*Katzman, 1986*), all mutations in *APP* resulted in early-onset AD or cerebral amyloid angiopathy. The previous exclusive linkage of APP to the rarer early-onset forms of AD, combined with the failures of anti-amyloid clinical trials to improve cognition in late-onset AD patients, led some to suggest separate disease etiologies in the late and early onset forms of AD. That A673T *APP* protects against late-onset AD would argue against this separation and suggest that intervention in APP processing is sufficient for AD prevention, as long as these interventions correct pathogenic alterations in APP metabolism without inhibiting APP-cleaving enzymes, which have multiple substrates.

The second notable aspect of the A673T mutation is that it does not per se exclusively implicate Aβ in AD pathogenesis. As noted in the initial report (*Jonsson et al., 2012*), examined in follow-up studies (*Maloney et al., 2014*), and demonstrated here for the first time in an animal KI model, the A673T mutation, via a decrease in affinity in APP for BACE1, results in, as a secondary effect, an attenuation of Aβ production. However, the direct consequences of this mutation are a decrease in βCTF and sAPPβ production, the metabolites generated by β-cleavage of APP. In addition, α-cleavage of the protective APPp mutant is increased, causing increased production of αCTF and sAPPα and possibly, as a secondary effect, of P3 peptides. Changes in APP metabolism caused by the protective Icelandic mutation are not as dramatic in magnitude as those induced by the pathogenic Swedish mutation. As a consequence, some variability in data significance between different animal cohorts and different experiments emerges. This variability is not surprising taking into consideration the inherent and inevitable differences between biological samples and the variability introduced by multistep biochemical analyses. In addition, as it was noted during the review process, 'the changes in APP metabolism, if observed, are in general more pronounced in female vs male rats'. This potentially important sex-difference will need to be fully explored in future studies with greater power.

Thus, in addition to Aβ many other metabolites of APP are differentially affected by the A673T mutation. Any of these metabolic changes, alone or in any possible combination, may mediate the protective effects of the Icelandic mutation. The evidence that the protective and pathogenic Swedish mutations have opposite effects on APP processing by β- and α-secretases, that is decreased β-cleavage and increased α-cleavage for the former and increased β-cleavage and decreased α-cleavage for the latter, is striking and may suggest that protection from and pathogenesis of dementia may depend upon a complex alteration in APP metabolites and their functions rather than simply on Aβ levels. These pathogenic/protective changes in APP metabolism are probably present during

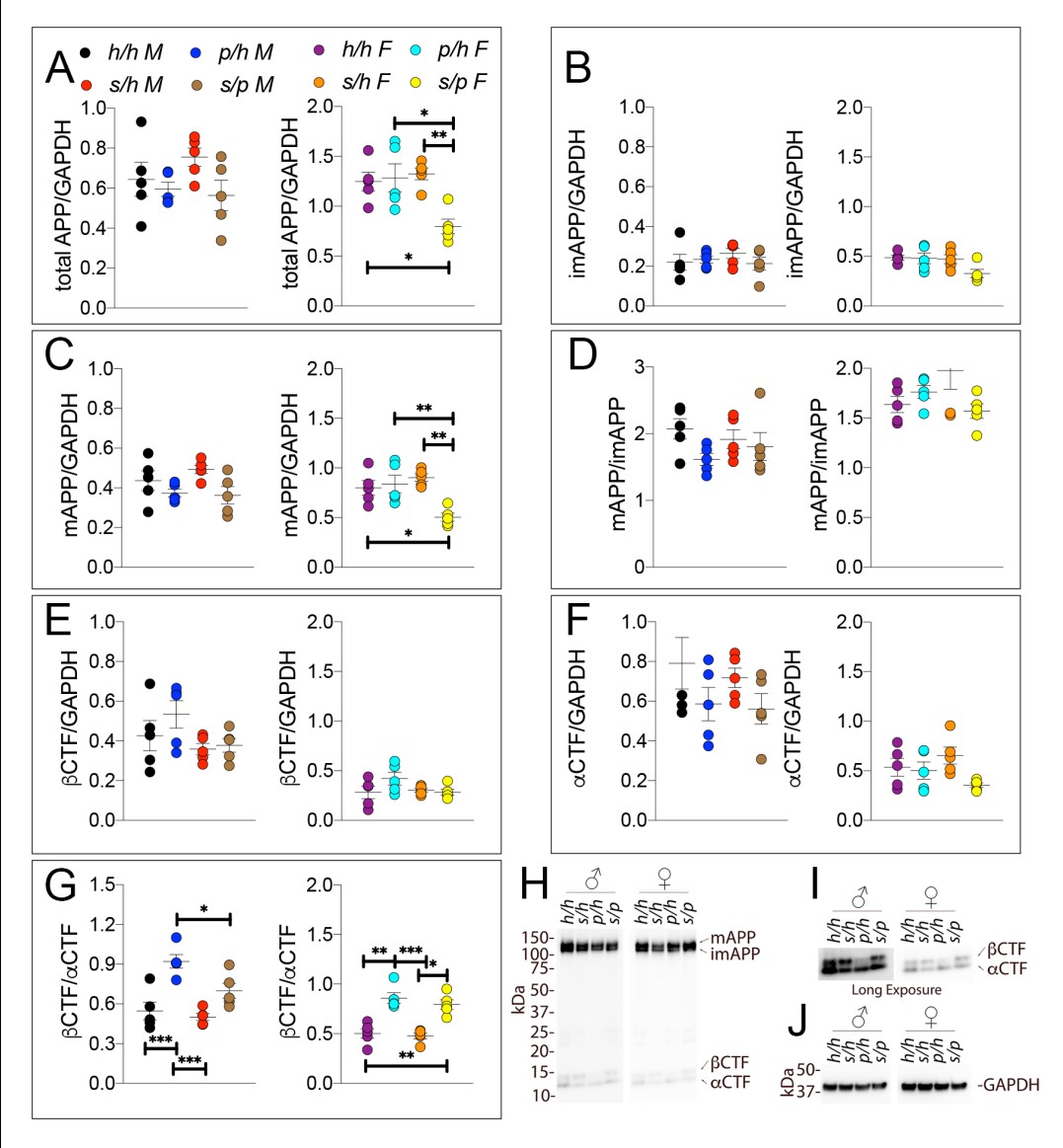

**Figure 10.** Western analysis of APP metabolites in $App^{h/h}$, $App^{s/h}$, $App^{p/h}$, and $App^{s/p}$ rats. Quantitation of WB analysis (A–G) with representative blots (H–J). (A) Normalized total APP levels in $App^{h/h}$, $App^{s/h}$, $App^{p/h}$, and $App^{s/p}$ male (left) and female (right) rats. (B) Normalized imAPP levels in $App^{h/h}$, $App^{s/h}$, $App^{p/h}$, and $App^{s/p}$ male (left) and female (right) rats. (C) Normalized mAPP levels $App^{h/h}$, $App^{s/h}$, $App^{p/h}$, and $App^{s/p}$ male (left) and female (right) rats. (D) Ratio of mAPP:imAPP in $App^{h/h}$, $App^{s/h}$, $App^{p/h}$, and $App^{s/p}$ rats. (E) Normalized βCTF levels in $App^{h/h}$, $App^{s/h}$, $App^{p/h}$, and $App^{s/p}$ male (left) and female (right) rats. (F) Normalized αCTF levels in $App^{h/h}$, $App^{s/h}$, $App^{p/h}$, and $App^{s/p}$ male (left) and female (right) rats. (G) Ratio of βCTF: αCTF. (H) Representative blot against C-terminus of APP (I) Longer exposure of α-APP-C-terminus blot to detect αCTF and βCTFs. (J) Anti-GAPDH loading control. Data are represented as mean ± SEM. Data were analyzed by Ordinary one-way ANOVA followed by post-hoc Tukey's multiple comparisons test when ANOVA showed statistically significant differences (*p<0.05; **p<0.01; ***p<0.001). Animals were analyzed at p28. We used 5 male and 5 female rats for each genotype. The online version of this article includes the following source data and figure supplement(s) for figure 10:

**Source data 1.** Related to *Figure 10A,B,C,D,E,F,G*.

**Figure supplement 1.** Western blots images used for quantitation shown in *Figure 10*.

embryonic development and the lifespan of carriers. Thus, even subtle changes may have a significant role in either pathogenic or protective mechanism, especially for diseases that manifest clinically at advanced age.

The fundamental differences between transgenic and KI approaches are illustrated by animal models of Familial Danish and British dementia (FDD and FBD). Overexpression of mutant genes produced an increase in Bri2 (the protein coded by *ITM2b*, the gene mutated in FDD and FBD) levels and a gain of function (*Coomaraswamy et al., 2010*; *Garringer et al., 2010*). In contrast, FDD and FBD KI mouse models showed reduction of Bri2 levels and function (*Tamayev et al., 2010a*; *Tamayev et al., 2010b*). Thus, FDD and FBD may be correctly modeled by either transgenic mice, which show a gain of function, or KI mice, which show a loss of function, but not by both. Paradoxically, either transgenic or KI mice may model the opposite of the pathogenic mechanisms causing FDD/FBD. Hypotheses of pathogenic mechanisms generated from incorrect model organisms may produce harm when they inform drug discovery and human clinical trials. In conclusion, transgenic and knock-in models may, for some diseases, be genetically distinct but valid disease models. For other diseases however, a transgenic approach may lead to incorrect and harmful assessments, which is unlikely to happen when using KI model organisms.

In addition to the quantity of these metabolites, there is a potential qualitative effect: the amino acid substitutions A673T alters the primary structure of several APP metabolites including full-length APP, Aβ, βCTF and sAPPα. These primary structure changes may, per se, have a protective effect. Reports of decreased tendency towards aggregation (*Benilova et al., 2014*) of A2T Aβ, the form of Aβ generated from A673T APP, suggest that the biological functions of other APP metabolites that bear the same mutation may also be affected. The biochemical effects caused by the protective *APP* mutation, therefore, are complex and can be exerted across multiple APP metabolites.

Processing of APP may also downregulate the function of the full-length precursor protein, with potential physiological and pathological consequences. For instance, β-secretase cleaves APP within a functional domain of APP called ISVAID, which interacts with glutamatergic synaptic vesicle proteins (*Yao et al., 2019*). Increased cleavage of APP by BACE1, as in *App$^s$* KI rats, disables the ISVAID and facilitates glutamate release (*Tambini et al., 2019*), a phenomenon known as BACE1 on APP-dependent glutamate release (BAD-Glu). It is possible that reduced β-cleavage of APP may lead to an opposite glutamatergic transmission alteration in *App$^P$* rats.

The generation of a KI-rat model of the protective *App* mutation has other advantages apart from the avoidance of the confounding effects of the transgenic approach. The protective model will be useful in testing what other AD-causing mutations and AD-related polymorphisms have on disease pathogenesis. Specifically, if a mutation acts via alteration in APP processing, theoretically, it would be possible to cross rat models with *App$^P$* rats to determine to what extent the mutation's or polymorphism's effects are APP-β/α-processing-dependent. The *App$^{s/P}$* rats are an example of such combined-mutation model organism. In *App$^{s/P}$* rats, which carry one protective and one pathogenic Swedish allele, the steady-state levels of APP metabolites are dictated by an additive effect of the two mutations. In this particular model, the effects of the Swedish mutation on APP metabolism, which are quantitatively greater as compared to those of the Icelandic variant, predominate. Thus, it is not likely that the *App$^P$* allele will protect from neurodegeneration caused by the Swedish allele, albeit this remains to be experimentally tested. However, pathological conditions leading to milder APP processing alteration may be counteracted by the *App$^P$* allele.

Just as the *App$^P$* KI-rat model can be used to determine interactions of other mutations on APP processing, it could also be used to determine if the mechanism of action of potential pharmacological AD treatments is β/α-processing dependent. For example, if the mechanism of action of a potential AD drug required a shift from β- to α-processing of APP, the effect would be attenuated in *App$^P$* rats.

Finally, the evidence that the Icelandic mutation delays/protects from what is known as aging-dependent normal cognitive decline (*Jonsson et al., 2012*) suggests that *App$^P$* rats will also be useful to understand mechanisms causing cognitive decline during aging. Mimicking these mechanisms pharmacologically may provide powerful tools to offset cognitive aging.

# Materials and methods

## Key resources table

| Reagent type (species) or resource | Designation | Source or reference | Identifiers | Additional information |
|---|---|---|---|---|
| Gene *App* (*Rattus norvegicus*) | *App*, Amyloid beta (A4) precursor protein | http://mar2016.archive.ensembl.org/Rattus_norvegicus/Gene/Summary?g=ENSRNOG00000006997;r=11:24425005-24641858 | Location: Chromosome 11 : 24,425,005–24,641,858 | Mutations were inserted into Exon 16 |
| Genetic reagent (*Rattus norvegicus*) | $App^h$ | (*Tambini et al., 2019*) | | Rat *App* allele with humanize Aβ region |
| Genetic reagent (*Rattus norvegicus*) | $App^s$ | (*Tambini et al., 2019*) | | Rat *App* allele with humanize Aβ region and pathogenic Swedish mutations |
| Genetic reagent (*Rattus norvegicus*) | $App^p$ | This paper | | Rat *App* allele with humanize Aβ region and protective Icelandic mutation |
| Recombinant DNA reagent | gRNA1 targeting vector | This paper: http://www.vectorbuilder.com/design/report/733e7a50-b705-45cf-8f8c-8b8206e6f174 | | Vector expressing gRNA1 |
| Recombinant DNA reagent | gRNA2 targeting vector | This paper http://www.vectorbuilder.com/design/report/2a73df2c-1d5c-4e69-b43a-ee6cfaea80c6 | | Vector expressing gRNA1 |
| Antibody | Y188 anti APP-C terminus (Rabbit polyclonal) | Abcam Cat# ab32136 | RRID:AB_2289606 | WB (1:1000) |
| Antibody | 6E10 anti human APP-Aβ$_{1-16}$, (Mouse monoclonal) | BioLegend Cat# 803001 | RRID:AB_2564653 | WB (1:1000) |
| Antibody | M3.2 anti rodent APP-Aβ$_{1-16}$, (Mouse monoclonal) | Biolegend Cat#: 805701 | RRID:AB_2564982 | WB (1:1000) |
| Antibody | Anti-sAPPα C-terminus, (Mouse monoclonal) | IBL Cat# 2B3 | | WB (1:1000) |
| Antibody | Anti-human sAPPβ WT C-terminus (Rabbit polyclonal) | Covance Cat# SIG-39138 | RRID:AB_662870 | WB (1:1000) |
| Antibody | Anti-human sAPPβ Sw C-terminus (Mouse monoclonal) | IBL Cat#: 6A1 | | WB (1:1000) |
| Antibody | Anti-GAPDH (Rabbit polyclonal) | Abcam Cat# ab32136 | RRID:AB_2289606 | WB (1:1000) |
| Antibody | HRP-Anti-mouse antibodies | Southern Biotech Cat#: 1031–05 | RRID:AB_2794307 | WB (1:1000) |
| Antibody | HRP-Anti-rabbit antibodies | Southern Biotech Cat# 4050–05 | RRID:AB_2795955 | WB (1:1000) |
| Antibody | HRP-Anti-rabbit antibodies | Cell Signaling Cat# 7074 | RRID:AB_2099233 | WB (1:1000) |
| Commercial assay or kit | West Dura ECL | Thermo | Cat# PI34076 | |

*Continued on next page*

*Continued*

| Reagent type (species) or resource | Designation | Source or reference | Identifiers | Additional information |
|---|---|---|---|---|
| Commercial assay or kit | Human β Amyloid (1-42) ELISA Kit – High Sensitive | Wako | | |
| Commercial assay or kit | Human β Amyloid (1-40) ELISA Kit II | Wako | | |
| Commercial assay or kit | V-PLEX Plus Aβ Peptide Panel 1 | Meso Scale Discovery | Cat# K15200G | |
| Commercial assay or kit | V-PLEX Plus Aβ Peptide Panel 1 4G8 | Meso Scale Discovery | Cat# K15199G | |
| Commercial assay or kit | sAPPβ-Sw Elisa), | Meso Scale Discovery | Cat# K151BUE | |
| Commercial assay or kit | WT sAPPα/sAPPβ | Meso Scale Discovery | Cat# K15120E | |
| Other | ChemiDoc MP Imaging System | Biorad | | |
| Instrument/ software, algorithm | MESO QuickPlex SQ 120 | Meso Scale Discovery | | |
| Other | xMark Spectrophotometer | Biorad | | |
| Instrument/ software, algorithm | Image Lab software | Biorad | RRID:SCR_014210 | |
| Instrument/ software, algorithm | GraphPad Prism | | RRID:SCR_002798 | |

## Generation of App KI rats

Generation of $App^{\delta 7}$, $App^s$, and $App^h$ rats was as described previously (*Tambini et al., 2019*). For details, see *Supplementary file 1*.

## Rats and ethics statement

All experiments were done according to policies on the care and use of laboratory animals of the Ethical Guidelines for Treatment of Laboratory Animals of the NIH. The procedures were described and approved by the Rutgers Institutional Animal Care and Use Committee (IACUC) (protocol number 201702513). All efforts were made to minimize animal suffering and reduce the number of animals used. The animals were housed two per cage under controlled laboratory conditions with a 12 hr dark light cycle, a temperature of $22 \pm 2°C$. Rats had free access to standard rodent diet and tap water.

## Rat brain preparation

Rats, 28 days old, were anesthetized with isoflurane and perfused via intracardiac catheterization with ice-cold PBS. Brains were extracted and homogenized using a glass-Teflon homogenizer (w/v = 100 mg tissue/1 ml buffer) in 20 mM Tris-base pH 7.4, 250 mM Sucrose, 1 mM EDTA, 1 mM EGTA plus phosphatase and protease inhibitors (ThermoScientific): all steps were carried out on ice or at 4°C. Total lysate was solubilized with 0.1% SDS and 1% NP-40 for 30 min rotating. Solubilized lysate was spun at 20,000 g for 10 m, the supernatant was collected and analyzed by ELISA and Western blotting. For *Figure 2D*, soluble lysate was prepared by the centrifugation of total lysate at 100,000 g for 30 m. The supernant was collected for further analysis.

## RT-PCR

RNA was extracted from P28 rat brain's using the RNeasy RNA Isolation kit (Qiagen 74104). cDNA was generated with a High-Capacity cDNA Reverse Transcription Kit (Thermo 4368814). Real-time

PCR was performed using 50 ng of cDNA, TaqMan Fast Advanced Master Mix (Thermo 4444556), and the appropriate TaqMan (Thermo) probes. Samples were analyzed on a QuantStudio 6 Flex Real-Time PCR System (Thermo). LinRegPCR software (hartfaalcentrum.nl) was used to quantify relative mRNA amounts. The probe Rn00570673_m1 (exon junctions 11–12, 12–13, and 13–14) was used to detect rat *App* mRNA. Samples were normalized to Gapdh mRNA levels detected with Rn01775763_g1 (exon junctions 2–3, and 7–8).

## Western analysis

Protein content was quantified by Bradford analysis prior to solubilization. 15 µg of protein plus LDS Sample buffer and 10% β-mercaptoethanol (Invitrogen NP0007) were separate by PAGE on a 4–12% Bis-Tris polyacrylamide gel (Biorad 3450125), transferred onto nitrocellulose using the Trans-blot Turbo system (Biorad) and visualized by red Ponceau staining. After membranes were blocked in 5%-milk (Biorad 1706404), the following primary antibodies were applied (overnight at 4˚C, at 1:1000 dilution in blocking solution (Thermo 37573): Y188 (Abcam ab32136), 6E10 (BioLegend 803001), sAPPα (IBL 2B3), M3.2 (Biolegend 805701), sAPPβ (Covance Catalog# SIG-39138), sAPPβ-Sw (IBL 6A1), and GAPDH (Sigma G9545). After extensive washings in PBS/Tween20 0.05%, the following secondary antibodies were used diluted 1:1000 in 5%-milk: anti-mouse (Southern Biotech, 1031–05) and a 1:1 mix of anti-rabbit (Southern Biotech, 4050–05) and anti-rabbit (Cell Signaling, 7074). Secondary antibodies were incubated with membranes for 30 min, RT, with shaking). After extensive washings in PBS/Tween20 0.05%, blots were developed with West Dura ECL reagent (Thermo, PI34076), visualized with ChemiDoc MP Imaging System (Biorad) and signal intensities were quantified with Image Lab software (Biorad).

## ELISA

Initial measurements of Aβ40 and Aβ42 content of 100 µg total solubilized rat brain lysate was done with Human β Amyloid (1-42) ELISA Kit – High Sensitive (Wako) and Human β Amyloid (1-40) ELISA Kit II (Wako). Absorbances at 450 nm were read on an xMark Spectrophotometer (Biorad).

For analysis of Aβ and sAPPs, the following Meso Scale Discovery kits were used: Aβ38, Aβ40, and Aβ42 were measured with V-PLEX Plus Aβ Peptide Panel 1 6E10 (K15200G) and V-PLEX Plus Aβ Peptide Panel 1 4G8 (K15199G), sAPPβ-Sw was measured with sAPP Swedish sAPPβ (K151BUE), and sAPPα/β-WT were measured with sAPPα/sAPPβ (K15120E), according to the manufacturer's recommendations. Plates were read on a MESO QuickPlex SQ 120. Data were analyzed using Prism software and represented as mean ± SEM.

## Statistical analysis

Data were analyzed using GraphPad Prism software and expressed as mean ± s.e.m. Differences between two groups were assessed by appropriate two-tailed unpaired Student's t-test Differences among three or more groups were assessed by One-way ANOVA. Data showing statistical significance by one-way ANOVA were subsequently analyzed by Tukey's multiple comparisons post hoc test. $p < 0.05$ was considered statistically significant. Statistical data are shown in the Figures.

---

# Additional information

### Funding

| Funder | Grant reference number | Author |
| --- | --- | --- |
| National Institute on Aging | R01AG063407 | Luciano D'Adamio |
| National Institute on Aging | RF1AG064821 | Luciano D'Adamio |

The funders had no role in study design, data collection and interpretation, or the decision to submit the work for publication.

### Author contributions

Marc D Tambini, Conceptualization, Data curation, Formal analysis, Validation, Investigation, Methodology; Kelly A Norris, Validation, Investigation; Luciano D'Adamio, Conceptualization, Resources,

Data curation, Formal analysis, Supervision, Funding acquisition, Validation, Investigation, Methodology, Project administration

## Author ORCIDs
Marc D Tambini ⬤ https://orcid.org/0000-0003-4461-586X
Luciano D'Adamio ⬤ https://orcid.org/0000-0002-2204-9441

## Ethics
Animal experimentation: Rats were handled according to the Ethical Guidelines for Treatment of Laboratory Animals of the NIH. The procedures were described and approved by the Institutional Animal Care and Use Committee (IACUC) at Rutgers University.(protocol number 201702513).

## Decision letter and Author response
Decision letter https://doi.org/10.7554/eLife.52612.sa1
Author response https://doi.org/10.7554/eLife.52612.sa2

## Additional files
### Supplementary files
• Supplementary file 1. Generation of App KI rats.

• Transparent reporting form

### Data availability
All data generated and analyzed are included. Source files have been provided for all Figures.

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
