## [Decision Letter]

**Acceptance summary:**

The study convincingly demonstrates that the Icelandic APP mutation that protects from sporadic Alzheimer's disease has the opposite effect on APP processing compared to the Swedish pathogenic APP mutation, emphasizing the role of APP and its metabolism in Alzheimer's disease pathogenesis. In addition, the rat knockin models demonstrate clear advantages over previously generated APP models.

**Decision letter after peer review:**

Thank you for resubmitting your work entitled "Opposite changes in APP processing and human aβ levels in rats carrying either a protective or a pathogenic APP mutation" for further consideration by *eLife*. Your revised article has been evaluated by Andrew West (Reviewing editor) and Huda Zoghbi (Senior Editor), as well as three reviewers working in the field.

There are three issues we would like addressed before acceptance:

1) All three reviewers found distracting grammatical errors and typos, especially in the Introduction, as well as some under-referenced sections. I would like you to focus on fixing these issues fully, some of which have been highlighted by the reviewers below. Further there were a few points made about clarity of the statistics that need attention.

2) In the spirit of full open access, there may be some points of similarity between your past 2019 Aging Cell paper and this current manuscript. However, please be sure to avoid any copyright issues and verbatim copying of text/figures from that past manuscript is not going to be permitted. Further, a few points of clarification are needing in the existing figure set, possible labeling problems or non-representative gels shown. These are outlined below.

3) After some internal discussions, no new data are needed for acceptance of your manuscript. However, if you have generated data to address some of reviewer 3's points that you may have generated in the time frame since initial submission, please feel free to include.

Reviewer #1:

In spite of many attempts, a good murine model of the APP-dependent pathology of Alzheimer's disease remain elusive. In the study described here, a rat knockin model of the Icelandic APP mutation was developed and characterized with multiple advantages over previously generated models. This model is extensively analyzed for the effect of the mutation on APP processing and the data compared to the changes in a previously generated and characterized rat knockin model expressing APP with the Swedish mutation. The importance of this work is that the data show that the Icelandic APP mutation that protects from sporadic Alzheimer's disease has the opposite effect on APP processing compared to the Swedish pathogenic APP mutation, emphasizing the role of APP and its metabolism in Alzheimer's disease pathogenesis.

Reviewer #2:

In this manuscript by Tambini, Norris, and D'Adamio, the authors aim to investigate APP protective and pathogenic mechanisms through their generated *App^p^* and *App^s^* knock-in (KI) rats. Tambini et al. developed this KI rat model with the background knowledge that cleavage of APP by α-secretase initiates a non-amyloidogenic pathway, while the β-secretase pathway leads to amyloid-β production, thereby opposing the α-secretase APP cleavage pathway. Moreover, in familial Alzheimer's disease (AD), the Icelandic APP mutation that cleaves at the BACE1 site is protective against developing sporadic dementia. Due to the nature of the Swedish mutation, *App^s^*, and the protective mutation, *App^p^*, in rats, Tambini et al. concluded that AD pathogenesis depends on a combination, as well as reverse modifications, of APP metabolism than the focusing of only on amyloid-β levels. The novelty of the study comes with the use of CRISPR/Cas9-mediated genome editing to develop the KI rats with both the protective and pathogenic APP mutations. Additionally, the KI mutation was chosen to be done in rats over the typical mouse models as the authors cite numerous reasons for the rat species being better suited to study neurodegenerative diseases than mice. The authors conclude that, since that theses rat models will be helpful to the neurodegeneration field in being able to understand aging cognitive decline mechanisms to allow for (eventual) pharmacological intervention. Overall, while a novel rat model, to be sure, could be of great benefit to the AD basic and translational fields, this manuscript does have concerns that would need to be addressed prior to being accepted by *eLife*.

Concerns that need to be addressed:

1) One large issue with the manuscript is the continual references to Tambini et al., 2019 in Aging Cell earlier this year. Figure 2 A-D in this submitted manuscript are directly replicated from the 2019 Aging Cell published article. Is it necessary to repeat the entire generation of the KI rat model in this manuscript as it was already done in another article by the same authors? Additionally, to be perfectly honest, this manuscript looks as if it should have been published first, before the Aging Cell article, as the "Facilitation of glutamate, but not GABA, […]" builds off developing the rat KI model.

2) In the Introduction and Discussion, the abbreviations of Familial Danish and British dementia are found to be swapped incorrectly.

3) In the Introduction paragraph describing analysis of transgenic APP models being confounded by several factors, it would be preferential to have these reasons cited to add validity to the authors' argument.

4) The Introduction has numerous grammatical errors and spacing next to parenthesis that needs to be addressed.

5) All figures should have asterisks within the bar graphs denoting statistical significance (if present), with the p-values being cited in the main text rather than in the tables underneath the bar graphs.

6) Why does Figure 2B's western blot looks so different compared to Figure 2A, C andD? Presumably these were all run on the same machine, which appears to be a capillary electrophoresis instrument; yet, Figure 2B displays lighter contrast than the other blots and does not seem to be in the linear portion of the curve like the others. Also, there is a 'smile' that occurs in a western blot gel, but not in capillary electrophoresis.

Reviewer #3:

Tambini et al. report the generation of a new *App^p^* knock-in rat model carrying a protective Icelandic mutation in the *App* gene, and the characterization of the APP metabolism in the newly generated model. In addition, they compare APP metabolism in the novel model vs. rats carrying the pathogenic APP Swedish mutation. Considering the limitations of the widely used "overexpression" animal models in the Alzheimer's disease (AD) field, the generation of knock-in models is of critical importance. Furthermore, as mentioned and discussed in the manuscript, the generated rat model present several advantages over the mouse ones for memory and cognitive research. These points support the relevance of the presented studies. Nevertheless, the manuscript is largely descriptive and the derived insights are not entirely novel but rather incremental. In this regard, the study presents several intriguing, sex-dependent phenotypes that could offer novel insights, however they remain as observations.

1) In order to demonstrate the relevance of the novel rat model for AD, analysis of the AD pathological hallmarks as well as behavioural changes should be performed.

2) In the Figure 2, the authors analyse the APP proteolysis in *App^p/p^* rats. As stated in the text, they could have not accurately measured the βCTF levels using Y188 antibody. Nevertheless, they use this antibody again in Figures 4 and 6. This reviewer suggests including only data generated with the validated 6E10 antibody.

3) In several figures, the authors show very complex statistical comparisons (all the conditions are compared to each other) but little is discussed about the relevance of the observations. For instance, it is interesting that the changes, if observed, are in general more pronounced in female vs. male rats, but the observations are not discussed. The complex presentation and lack of discussion only leads to the loss of the main message.

4) There is an overlap between data presented in Figures 3 and 5 and 7. However, the overlapping data seem to be inconsistent between the figures. For example, trends towards decreased sAPPα levels are seen in *App^p/h^* M vs. *App^h/h^* M rats in Figure 7 while an opposite trend is seen in Figure 3. Similarly, there is an overlap between Figure 4, 6 and 9. The overlapping data seem to be inconsistent between the figures for the βCTF/αCTF analyses.

5) In the Introduction, the authors write that the Icelandic mutation increases the rate of APP cleavage by α-secretase. However, this aspect is uncertain. In the original, cited publication (Jonsson et al., 2012) "sAPPα trended non-significantly towards an increase".

---

## [Author Response]

There are three issues we would like addressed before acceptance:1) All three reviewers found distracting grammatical errors and typos, especially in the Introduction, as well as some under-referenced sections. I would like you to focus on fixing these issues fully, some of which have been highlighted by the reviewers below. Further there were a few points made about clarity of the statistics that need attention.

We are very sorry for the mistakes and thank the reviewers for noticing them. The mistakes have been corrected. In addition, all figures of the revised manuscript have asterisks within the bar graphs denoting statistical significance when present. Statistical data are cited in the main text.

2) In the spirit of full open access, there may be some points of similarity between your past 2019 Aging Cell paper and this current manuscript. However, please be sure to avoid any copyright issues and verbatim copying of text/figures from that past manuscript is not going to be permitted. Further, a few points of clarification are needing in the existing figure set, possible labeling problems or non-representative gels shown. These are outlined below.

We have edited the text/figures to avoid copyright issues. The problems with figures have been corrected.

3) After some internal discussions, no new data are needed for acceptance of your manuscript. However, if you have generated data to address some of reviewer 3's points that you may have generated in the time frame since initial submission, please feel free to include.

After reading the criticisms, we have decided to perform a part of the new experiments suggested, namely the biochemical analysis with 6E10. These new data are included in a new figure (Figure 7). Therefore, the previous Figures 7, 8 and 9 are, in the revised paper, Figures 8, 9 and 10, respectively.

Reviewer #2:[…]Concerns that need to be addressed:1) One large issue with the manuscript is the continual references to Tambini et al., 2019 in Aging Cell earlier this year. Figure 2 A-D in this submitted manuscript are directly replicated from the 2019 Aging Cell published article. Is it necessary to repeat the entire generation of the KI rat model in this manuscript as it was already done in another article by the same authors? Additionally, to be perfectly honest, this manuscript looks as if it should have been published first, before the Aging Cell article, as the "Facilitation of glutamate, but not GABA, […]" builds off developing the rat KI model.

We prefer to include this section since the generation of *App^p^* rats and off-target analysis was not previously described. The Aging Cell paper described the generation of *App^h^* and *App^s^* rats, but not *App^p^* rats.

2) In the Introduction and Discussion, the abbreviations of Familial Danish and British dementia are found to be swapped incorrectly.

Sorry for the mistakes and thank you for noticing them. The mistakes have been corrected.

3) In the Introduction paragraph describing analysis of transgenic APP models being confounded by several factors, it would be preferential to have these reasons cited to add validity to the authors' argument.

Citations have been added.

4) The Introduction has numerous grammatical errors and spacing next to parenthesis that needs to be addressed.

Sorry for the mistakes and thank you for noticing them. The mistakes have been corrected.

5) All figures should have asterisks within the bar graphs denoting statistical significance (if present), with the p-values being cited in the main text rather than in the tables underneath the bar graphs.

All figures of the revised manuscript have asterisks within the bar graphs denoting statistical significance when present. Statistical data are cited in the main text.

6) Why does Figure 2B's western blot looks so different compared to Figure 2A, C and D? Presumably these were all run on the same machine, which appears to be a capillary electrophoresis instrument; yet, Figure 2B displays lighter contrast than the other blots and does not seem to be in the linear portion of the curve like the others. Also, there is a 'smile' that occurs in a western blot gel, but not in capillary electrophoresis.

All western blots were run on BioRad’s Criterion system, which uses traditional polyacrylamide gels, precast in our case. All samples were prepared in loading buffer and loaded onto several replicate gels which were each blotted with a different antibody. We run the gels in the same buffer and at the same voltage, but sometimes the gels “smile.” We suspect, given that these are precast gels, there may be some differences across batches. Additionally, our machines hold 2 gels per apparatus, and the differences described may reflect that they were run on different apparatuses. We avoid stripping and reblotting as this may introduce artifacts (protein effaced from gel, high background, nonspecific bands from prior blots).

Reviewer #3:[…]1) In order to demonstrate the relevance of the novel rat model for AD, analysis of the AD pathological hallmarks as well as behavioural changes should be performed.

Those are indeed important questions that we are addressing in long-term longitudinal studies. However, the present study’s goal is to address the early impact of these APP mutations on APP metabolism. Thus, pathological and behavioural studies are outside the scope of the present study.

2) In the Figure 2, the authors analyse the APP proteolysis in App^p/p^ rats. As stated in the text, they could have not accurately measured the βCTF levels using Y188 antibody. Nevertheless, they use this antibody again in Figures 4 and 6. This reviewer suggests including only data generated with the validated 6E10 antibody.

We have performed these experiments that are shown in the new Figure 7 (and in Figure 7—figure supplement 1). Because of this addition, the previous Figures 7, 8 and 9 are now shown as Figure 8, 9 and 10, respectively. The previous figure supplement S3 is now Figure 10—figure supplement 1.

3) In several figures, the authors show very complex statistical comparisons (all the conditions are compared to each other) but little is discussed about the relevance of the observations. For instance, it is interesting that the changes, if observed, are in general more pronounced in female vs. male rats, but the observations are not discussed. The complex presentation and lack of discussion only leads to the loss of the main message.

Thank you for noticing and pointing this out. We did notice that the changes in APP processing caused by the Swedish and Icelandic mutations seem to be clearer in females. However, we preferred, and prefer, not to discuss in depth potential sex-differences to avoid overinterpretation of our data. It is possible that analysis of much larger number of animals may indeed unveil sex-specific changes in APP metabolism caused by these mutations. But we have introduced the following sentence in the Discussion: “In addition, as it was noted during the review process, “the changes in APP metabolism, if observed, are in general more pronounced in female vs. male rats”. This is a potential interesting and important gender-difference that will need to be fully explored in future studies with greater power.”

4) There is an overlap between data presented in Figures 3 and 5 and 7. However, the overlapping data seem to be inconsistent between the figures. For example, trends towards decreased sAPPα levels are seen in App^p/h^ M vs. App^h/h^ M rats in Figure 7 while an opposite trend is seen in Figure 3. Similarly, there is an overlap between Figure 4, 6 and 9. The overlapping data seem to be inconsistent between the figures for the βCTF/αCTF analyses.

For the data in Figure 9 (now Figure 10), there was a mislabeling of the legend. *App^p/h^* and *App^s/h^* are switched in the quantitation section. The western blots to which they correspond are labeled correctly. Once the data are labeled correctly, the βCTF/αCTF analyses agree with the previous figures. We apologize for this error. For the data in Figure 7 (now Figure 8), that is correct, the discrepancy was also pointed out by reviewer 1 and has been addressed as follows. In Figure 3F, in which data were segregated by sex (n=5), there was a trend toward increase in sAPPα levels in *App^p/h^* vs. *App^h/h^* rats, for both males and females. The grouped data (n=10) in Figure 3G shows a significant increase. In Figure 7E (now Figure 8E), there is a trend seen only in females and not in males. When we look at pooled data from male and females there is no increase in sAPPα levels in *App^p/h^* rats (not shown). The discrepancy between these two sets of data are likely the result of inherent variability between biological samples, which may conceal a subtle change in α-processing of APP caused by the protective mutation. Accordingly, the Results section describing Figure 7E (now Figure 8E), data has been changed to acknowledge the differing results in Figures 3 and 7 (now Figure 8). Nevertheless, the data obtained with homozygous *App^p/p^* rats indicate an association of the protective mutation with reduced α-processing.

5) In the Introduction, the authors write that the Icelandic mutation increases the rate of APP cleavage by α-secretase. However, this aspect is uncertain. In the original, cited publication (Jonsson et al., 2012) "sAPPα trended non-significantly towards an increase".

We have changed the wording in the Introduction.